# The educational pathway to Advanced Practice for the physiotherapist: A systematic mixed studies review

**Kaitlyn Maddigan**[1,2*], **Chris Davis**[1,3], **Brendan Saville**[2], **Kathryn Nishimura**[2], **Jennifer Van Bussel**[1], **Andrews K. Tawiah**[1], **Katie L. Kowalski**[1], **Alison B. Rushton**[1]

1 School of Physical Therapy, Faculty of Health Sciences, Western University, London, Ontario, Canada,
2 Fowler Kennedy Sport Medicine Clinic, 3M Centre, Western University, London, Ontario, Canada,
3 Nuffield Health Learning Foundation, Nuffield Health, Surrey, England

* kmaddig@uwo.ca

## Abstract

### Background

Advanced Practice Physiotherapy (APP) is a higher level of practice grounded in 4 pillars: clinical practice, leadership, education and research. A critical step toward successful integration and sustainability of APP in healthcare systems is understanding the educational pathway to APP.

### Objectives

1) To describe the post-licensure educational pathways that physiotherapists engage in to advance their level of practice.
2) To evaluate demonstration of the pillars of APP by the physiotherapist after traversing a post-licensure educational pathway.

### Methods

This systematic mixed studies review is reported in accordance with PRISMA and pre-registered (PROSPERO: CRD42024499563). 8 databases plus the grey literature were searched. 2 independent reviewers determined eligibility, extracted data, assessed quality (QuADS) and determined the overall confidence in the cumulative evidence (GRADE-CERQual).

### Results

81 studies (18 qualitative, 17 mixed methods, 46 quantitative) were included in a data based convergent qualitative synthesis. 6 distinct post-licensure educational pathways were described and evaluated: Masters level education, residency and fellowship programs, accredited area of practice education, mentorship, multiple encounter courses and single encounter courses.

**Data availability statement:** All relevant data are within the article and its Supporting Information files.

**Funding:** The author(s) received no specific funding for this work.

**Competing interests:** The authors have declared that no competing interests exist.

## Conclusion

There is a high level of confidence (GRADE-CERQual) in the finding that Masters level education consistently resulted in all 4 pillars demonstrated by the physiotherapist. Masters level education appears to be the optimal pathway to APP.

---

## Introduction

### Rationale

Advanced Practice Physiotherapy (APP) is a higher level of practice, grounded in 4 pillars: clinical practice, leadership, education and research [1]. It necessitates distinctly increased skills, clinical reasoning and experience which leads to improved service outcomes, patient experiences and includes providing care to patients with complex needs, both safely and competently [1–11]. APP has been implemented in 14 countries worldwide across 16 areas of practice, though it has not yet achieved global recognition [11]. APPs are most frequently found in musculoskeletal (MSK) care (including outpatient orthopaedics and sports physiotherapy), neurology, cardiorespiratory, and paediatrics. Their roles vary by specialty and region but often include requesting diagnostic imaging, ordering blood tests, performing injections, and independently prescribing or de-prescribing medications. APPs also conduct orthopaedic triage, screening patients in emergency departments and those referred for surgical consultation [11]. The capacity for APP to alleviate burdened health care systems has been demonstrated in several systematic reviews. This body of literature advocates that physiotherapists (PTs) working in Advanced Practice roles, triage appropriately, deliver accurate diagnoses and largely improve access to care and treatment outcomes for a range of patients [3–9].

World Physiotherapy has acknowledged that there is no globally defined educational pathway (EP) for the PT to APP and most importantly that this is a vital gap to be filled [2,11,12]. In the UK, a leading country in the establishment of APP, the EP to Advanced Practice is a post-licensure (PL) Masters level qualification, with an alternative portfolio-based route recognized as equivalent, provided that the practitioner can demonstrate evidence of the competencies of all 4 pillars [1,13,14]. However, a recent global survey revealed that only half of the 112 member organizations of World Physiotherapy agree that PTs should demonstrate a set of defined competencies and possess a PL qualification such as a Master's degree or PhD to be considered for an APP role [11]. Moreover, a recent scoping review on APP examined education curricula and advocated for uniform standards, emphasizing that despite variations in APP roles within and across countries, standardized education remains feasible [15]. This prior literature offers scaffolding to build upon; however, a significant area recognized by these earlier works that remains unaddressed is the outward existence of a wide range of PL educational pathways (PL-EP) that may lead to APP. Lack of standardized PL-EPs is a key factor that contributes to the slow acknowledgment, growth and integration of these roles into healthcare systems worldwide [11,13,16–18].

A synthesis of the PL-EPs PTs are engaging in to advance their level of practice, and how well the outcomes of these pathways align with the pillars of Advanced Practice for the PT is currently lacking in the literature. Integrating this information has the potential to aid in establishing a standardized educational framework to APP and contribute to APP opportunities, enabling career progression and transferability of roles across settings and jurisdictions to create opportunity for national and international regulation [2,11,15,19]. Thus, a thorough understanding of the educational journey that PTs undergo to become Advanced Practitioners is an imperative step towards standardization of APP, global recognition and widespread implementation.

### Objectives

1) To describe the PL-EPs that PTs engage in to advance their level of practice.

2) To evaluate demonstration of the pillars of Advanced Practice by the PT after traversing a PL-EP.

## Materials and methods

This systematic mixed studies review (SMSR) was conducted in line with a pre-defined and published protocol [20], and registered with the International Prospective Register of Systematic Reviews (PROSPERO: CRD42024499563). There were no deviations from the protocol. This systematic review is reported in accordance with the Preferred Reporting Items for Systematic Review and Meta-Analysis (PRISMA) 2020 checklist [21] (S1 Table)

### Study design and researcher positionality

A post-positivist lens, backed by pragmatism underpinned this review. All authors are PTs working in clinical practice and/ or education within a post-secondary institution, all of whom have an interest in APP [20]. The exploration of PL-EPs to Advanced Practice for the PT demands an appreciation for both the objective realities and the practical considerations shaping the professional landscape, as such a mixed studies approach was chosen. An SMSR permitted all relevant and available studies on the topic to be retrieved for a comprehensive synthesis of evidence, capable of producing statements to guide decision-making and policy development given that systematic reviews are considered the gold-standard for evidence synthesis [22].

### Eligibility criteria

### Information sources

MEDLINE (Ovid), Embase, CINAHL, the Cochrane Library, Web of Science, PEDro, SportDiscus and ProQuest Education databases were searched from inception to 02/29/2024. Grey literature was searched through ProQuest Dissertations and Theses, trial registers (ClinicalTrials.gov and World Health Organization International Clinical Trials Registry Platform) and

**Table 1. Eligibility criteria informed by the PICOS framework [23].**

| Population | Physiotherapists (Physical Therapists) |
|---|---|
| Intervention | PL-EPs of any form but most commonly Masters level or post-graduate education, mentorship programs, residency programs, workshops or workplace training across all areas of practice. |
| Comparison | N/A |
| Outcomes | 1. description of PL-EPs<br>2. demonstration of any of the 4 pillars of Advanced Practice by the physiotherapist |
| Study Design | All primary research studies (qualitative, quantitative and mixed methods) |
| Publication Language | English, or papers able to be sufficiently translated into English via Google Translate |

Google (the first one hundred results were screened for inclusion). The references of included studies were screened to further supplement the search. Where full text studies could not be retrieved, access requests were made via email to the corresponding author.

## Search strategy

The search was constructed in collaboration with a Teaching and Learning Librarian at Western University, and was based on 3 concepts: the PT, PL-EPs and the competencies that underpin the 4 pillars of Advanced Practice [24]. KM carried out the searches independently using the search strategy initially developed in MEDLINE that was consistent across all databases, with specific search terms adjusted to reflect database appropriate syntax. Example MeSH terms used to search the PT concept were, Physical Therapists and Physical Therapy Modalities, for the PL-EP concept, Education-Graduate, Education-Professional, Mentoring, Inservice Training, Internship and Residency and for the pillars concept were, Clinical Competence, Professional Competence, Clinical Reasoning, Professional Role, Accreditation, Program Evaluation, Leadership, Communication, Research and Evidence Based Practice. Detailed search strategies, including an exhaustive list of search terms used are available in S2 Table.

## Selection process

Covidence [25], an internet-based collaboration platform was used to import all citations, remove duplicates and assist in the process of determining eligibility. Studies were assessed by 2 researchers (KM and either CD, KN or BS) at each screening stage [26]. If it was clear from the title and abstract that the content was not relevant to the objectives, the study was excluded. Full-text copies of potentially relevant studies were acquired and subsequently screened for inclusion. Studies included at title and abstract stage were excluded if a full-text publication was not available and could not be retrieved or was confirmed non-existent after contacting the corresponding author. At each stage if discrepancies existed regarding eligibility, they were discussed between reviewers. Consensus was achieved in each of these cases without requiring a third reviewer to mediate. One study included for full text review was originally published in German. Google Translate was used to convert the publication into English and it was then evaluated for accuracy by a native German speaking PT [27].

## Data collection process

Data were extracted from included studies by KM in parallel with CD, BS or KN into a standardised data extraction form. Data items included: author, year, country and method of data collection, sample size, study objective, design and setting, characteristics of the EP, characteristics of the physiotherapist and evidence of the competencies that underpin the 4 pillars [20]. The tool was piloted on 6 studies prior to continuing with data extraction of all studies, and discrepancies in extracted data were resolved through discussion.

## Quality assessment

The Quality Assessment for Diverse Studies (QuADS) was used to determine a quality rating for each included study [28]. This tool was suitable as it facilitates a pragmatic understanding of included studies and is designed to assess the quality of all methodological study designs. The QuADS permitted each researcher to reflect and consider components of the study from a substantive position, gauging the extent to which each criterion was met [28]. The QuADS tool demonstrates substantial inter-rater reliability (k = 0.66) [29], face and content validity for application in systematic reviews with mixed, or multi-methods health services research [28]. KM evaluated quality of each included study in parallel with CD, BS or KN. If there was disagreement, consensus on the rating was reached through discussion. The tool was piloted on 6 studies and discussed in a meeting to agree on application of the QuADS criteria [30].

## Data synthesis

Heterogeneity of study design is an inherent challenge faced in SMSRs. As such, a structured and robust approach was used to connect the stories and numbers into a practical understanding of the EP to APP [31]. The synthesis was carried out using a data based qualitative convergent synthesis, which involved extracting data from studies of mixed designs and synthesizing it to convey convergent results (Fig 1) [32]. Due to the integrated nature of this approach, data transformation was necessary. Described by Pluye and Hong (2014) this was accomplished through qualitative thematic analysis to 'qualitize' all quantitative data [31]. This involved interpreting numerical results, often from surveys, questionnaires, performance evaluations etc through an inductive coding process, transforming numbers into words that capture their underlying meaning. By contextualizing the data in this way, findings were operationalized into meaningful explanations. Thereafter, for each objective the extracted data were analyzed using qualitative synthesis methods. A clustered textual description was adopted to address the first objective, and a directed content analysis implemented to address the second [20].The results were subsequently converged and synthesized through joint display in a Heat map [32–34].

## Confidence in cumulative evidence

GRADE-CERQual evaluates review findings from a qualitative evidence synthesis, defined as an analytic output describing a phenomenon based on primary study data. Since GRADE-CERQual does not require primary research to

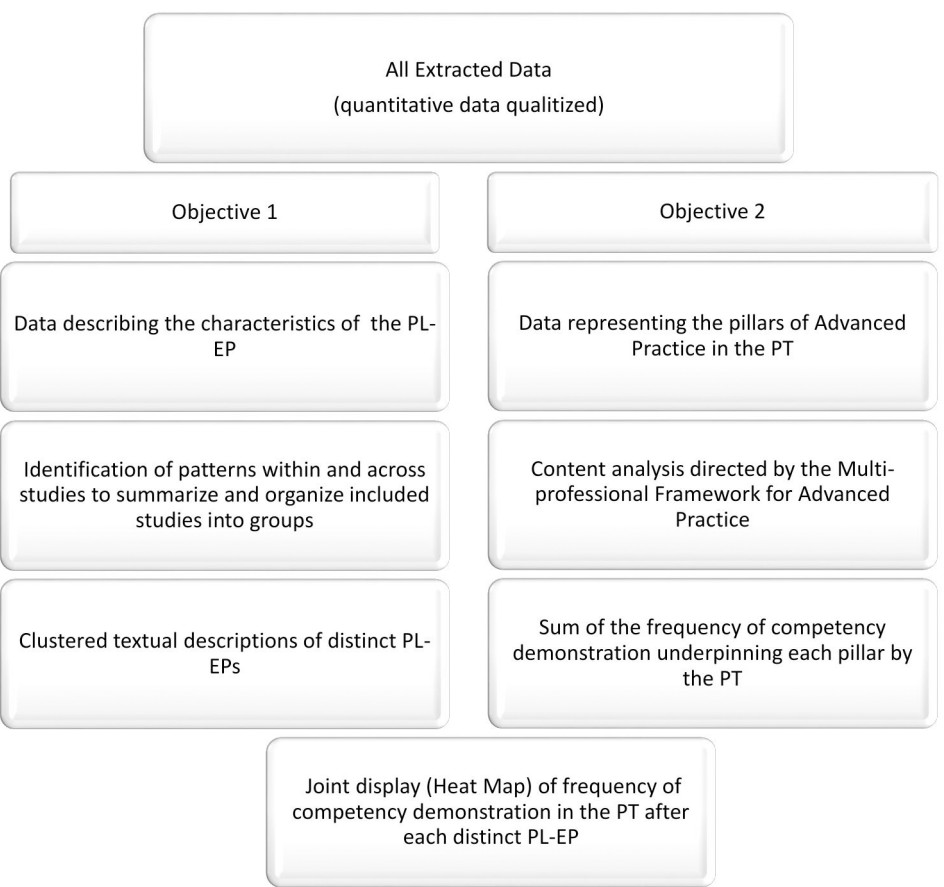

**Fig 1. Data based qualitative convergent synthesis approach [ 31–36].**

be qualitative and the synthesis in this review followed data transformation, this tool was appropriately used to assess confidence in the cumulative qualitative evidence synthesis [37,38].The 13 criteria of the QuADS informed the 4 categories of the CERQual when determining the level of confidence in each review finding. 9 of the QuADS criteria informed methodological limitations, 2 informed relevance and 2 informed adequacy of data along with a substantive assessment of the number of studies contributing to the finding [39–41]. The remaining CERQual component of coherence was assessed considering overall fit between the proposed review finding and the content from the primary studies [42]. Assessment was completed independently by KM and CD. Conflicts were resolved through discussion, not requiring a third reviewer. This tool provides a transparent, systematic framework to determine how much confidence to place in qualitative synthesis review findings ultimately increasing the usability of the findings [37].

## Results

### Study selection

The search strategy yielded 27,476 studies (Fig 2). After duplicate removal, 23,233 studies underwent title and abstract screening. 172 studies were screened at the full text stage. 81 studies met eligibility criteria for inclusion. At the full text stage there was substantial to almost perfect reliability between reviewers (K = 0.69–0.92) and after discussion there was 100% agreement (k = 1.0), with no need for third reviewer arbitration.

### Study characteristics

16 countries across 5 continents were represented in the 81 studies included in this review. Studies were published between 1987 and 2024 spanning a diverse set of methodological designs, 46 quantitative, 18 qualitative and 17 mixed methods. Within the included studies were 2 doctorate dissertations and 1 article originally published in German. A summary of the results from individual studies can be found in Table 2, with additional detail in S3 and S4 Tables.

### Quality assessment

A summary of the quality assessment score and converted percentage for each study using the QUADs is provided in Table 3 [30,124]. Study quality ranged 41–100%, with 58/81 studies scoring >75%. These individual scores enable the quality of a study to be considered alongside its results. 12 review findings, summated from the synthesized primary data were assessed using the GRADE-CERQual. The assessment of cumulative evidence determined there to be high confidence in 6/12, moderate confidence in 4/12 and low confidence in 2/12 review findings (S5 Table).

### Synthesis Findings

**Educational Pathways.** 6 distinct PL-EP that may be delivered across any area of practice were recognized in this review: Masters level education, residency and fellowship programs, accredited area of practice education, mentorship, multiple encounter courses and single encounter courses. Clustered textual descriptions of each pathway are presented below.

**Masters Level Education.** 9 studies investigated Masters level education [59,72,90,99,102,103,108,112,115]. This PL-EP is delivered over an extended time-period, often spanning 1–2 years. It is accredited, highly structured and held to international standards. In most instances it includes a component of mentorship, aimed at practical skill development and tends to be focused in the MSK area of practice [72,90,99,102,103,108,115]. Distinct to this pathway is the frequent inclusion of research skill development and focus on evidence informed practice. Further, it focuses on the development of critical thinking and analysis, intends to evoke self-reflection in the PT and necessitates formal student assessment (i.e., written or practical examination) prior to granting the academic award of a Masters degree.

**Residency and Fellowship Programs.** 16 studies investigated residency and fellowship programs [52,55,61–64,71,76,89,94,104,107,113,114,117,122]. This PL-EP extends 1–2 years in duration, is a structured and

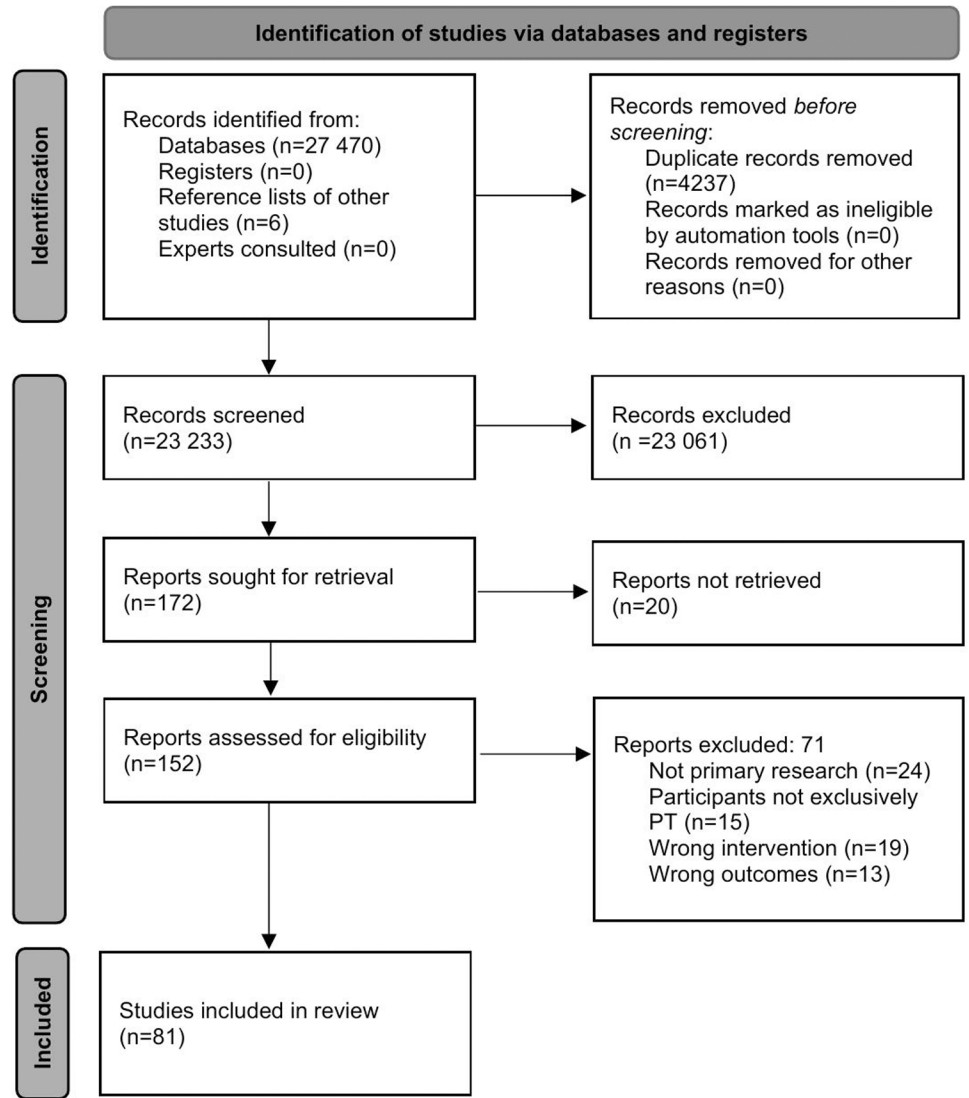

**Fig 2. PRISMA flow diagram [ 21].**

accredited route that is always focused to a specific area of practice, most commonly MSK. There is an emphasis on the development of evidence informed clinical reasoning and personal reflection. It's defined by measurable learning objectives and necessitates students to successfully perform on assessment prior to completion. A distinct feature of this pathway is its emphasis on experiential learning and mentored clinical practice.

**Accredited Area of Practice Education.** 4 studies investigated accredited area of practice education [45,47,67,106]. This PL-EP spans 10 months to 3 years [45,47]. Unique to this pathway is accreditation outside university infrastructure by a governing body such as the International Federation of Orthopedic Manual and Musculoskeletal Physical Therapists (IFOMPT), where there are established standards for competency. It results in the granting of a diploma or certificate upon successful completion of formal assessment. This pathway tends to be focused to the MSK area of practice and includes the development of hands-on manual therapy skills, achieved in classroom environments but also via clinical mentorship [45,47,67,106].

**Table 2. Results of individual studies.**

| Author Year | Country Setting | Methodology Study Design | Aim of Study | PL-EP | Duration (dur) Accredited (acrd) Area of Practice (aop) | Key Features of PL-EP | n age (avg +/- SD) exp (avg +/- SD) | Data Collect Tool | Pillars of Advanced Practice |
|---|---|---|---|---|---|---|---|---|---|
| Adhikari 2020 [43] | Nepal Private and Public Settings | Quantitative Pre-Post Design | to evaluate the effectiveness of an evidence-based structured educational workshop in enhancing PT's clinical decision-making skills | Single Encounter Course | dur: 6 hrs acrd: yes aop: non-specific | Didactic Approach Case Study Approach Student Assessment | n 42 (24 and 18) age 26.95 +/-3.58 exp: < 2 yrs 57.1%; ≥ 2 yrs 42.9% | Clinical Reasoning Ax | Clinical Practice |
| Allison 2023 [44] | Australia Online | Quantitative RCT | evaluate effects of an online education program about weight management for OA on PT's self-reported confidence in knowledge and skills in weight management, and attitudes toward obesity | Single Encounter Course | dur: 8 hrs acrd: no aop: MSK | Didactic Approach Case Study Approach Student Assessment | n: 80 (41 Ctrl, 39 Ed) age: not reported exp: Ctrl 14.6 yrs: Ed 11.9 yrs | Survey or Questionnaire | Clinical Practice |
| Anderseck 2020 [45] | Germany Online | Quantitative Survey Methodology | to explore whether an OMT educational programme (orthopedic manual/manipulative physiotherapy) increases employability, and which criteria indicate a relevant improvement | Accredited Area of Practice Training | dur: 260–1000 hrs acrd: yes aop: MSK | Skill Practice Student Assessment | n: 194 age: 25–65 yrs (range) exp: not reported | Survey or Questionnaire | Clinical Practice Education Research |
| Balogun 2018 [46] | Nigeria | Quantitative Pre-Post Design | to evaluate the impact of a three-hour, professionalism-focused, continuing education workshop on the knowledge and attributes of professionalism of physical therapists | Single Encounter Course | dur: 3 hrs acrd: no aop: non-specific | Didactic Approach Case Study Approach | n: 47 age: 41 +/- 10.1 yrs exp: 15 +/- 9.8 yrs | Survey or Questionnaire | Clinical Practice |
| Banks 2013 [47] | UK Hospital | Mixed Methods Pre-Post Design | to investigate further whether a structured in-service education programme (based on learning outcomes from assessment of feedback on clinical practice), along with clinical assistance sessions and protected self-directed learning time, contribute to measurable as well as perceived improvement in clinical competencies over an academic calendar year | Accredited Area of Practice Training | dur: 21 hrs in modules + 10 hrs mentorship acrd: yes aop: MSK | Didactic Approach Mentorship Case Study Approach Self Reflection Skill Practice | n: 20 age: not reported exp: not reported | Mentor or Instructor Evaluation Survey or Questionnaire | Clinical Practice Research |
| Barton 2021 [48] | Australia Public and private settings | Mixed Methods Pre-Post Design | to investigate changes in physiotherapists' practices, and confidence and beliefs about capabilities to provide patient education and exercise-therapy to people with knee osteoarthritis | Single Encounter Course | dur: 2 days acrd: no aop: MSK | Didactic Approach Case Study Approach Skill Practice | n: 1064 age: n/a exp: < 5 yrs 34%; 5–10 yrs 20%; 11–15 yrs 11%; > 15 yrs 36% | Survey or Questionnaire | Clinical Practice Leadership |

*(Continued)*

Table 2. (Continued)

| Author Year | Country Setting | Method-ology Study Design | Aim of Study | PL-EP | Duration (dur) Accredited (acrd) Area of Practice (aop) | Key Features of PL-EP | n age (avg +/- SD) exp (avg +/- SD) | Data Collect Tool | Pillars of Advanced Practice |
|---|---|---|---|---|---|---|---|---|---|
| Bastick 2020 [49] | Australia Tertiary teaching hospital and pub-lic health service | Mixed Methods Pre-Post Design | to investigate the effects of stream-specific clinical training on junior phys-iotherapist self-efficacy, self-rated confidence, and self-rated ability to work independently during weekend shifts | Multiple Encoun-ter Course | dur: 8 weeks acrd: no aop: non-specific | Didactic Approach Mentor-ship Skill Practice | n: 18 age: 26 yrs exp: 1/4–9 yrs (range) | Survey or Question-naire | Clinical Practice |
| Bird 2022 [50] | Australia Private Practice | Mixed Methods Pre-Post Design | to develop, implement, and evaluate custom-ized workshops for a PT private clinic based on the health literacy profile of the clinic's clients and evaluate the changes in health literacy knowledge and skills of physiother-apists working in the private clinic | Single Encoun-ter Course | dur: 5 hours acrd: no aop: non-specific | Didactic Approach Case Study Approach Skill Practice | n: 10 age: not reported exp: not reported | Semi-Structured Interviews Survey or Question-naire | Clinical Practice Leadership Education |
| Brennan 2006 [51] | USA Private Clinic | Quantita-tive Pre-Post Design | to examine the impact of a CE intervention provided to a group of physical therapists on the treatment of patients with neck pain and to deter-mine whether physical therapists who attended the CE course and par-ticipated in an ongoing clinical improvement project after completion of the course achieved more improvement in clinical outcomes than therapists who attended the CE course but did not participate in the clinical improvement project | Multiple Encoun-ter Course | dur: 2 days + 6 month follow up acrd: no aop: MSK | Didactic Approach Mentor-ship Skill Practice | n: 34 age: 40 yrs exp: 13.3 yrs | Patient Outcomes | Clinical Practice |
| Briggs 2023 [52] | USA Online | Quan-titative Survey Methodol-ogy | to compare perceived clinical competency and job duties between PTs with formal men-tored post-professional clinical education with PTs without formal post-professional clinical education. | Resi-dency and Fellow-ship | dur: not reported acrd: yes aop: non-specific | Mentor-ship Self Reflection | n: 2334 age: non-trained 45.3 +/-11.7 yrs, res-idency 34.4 +/-8.2 yrs, fellowship 42.6 +/-9.5 yrs exp: non trained 19.0 +/-12.1 yrs,res-idency 7.9 +/-8.2 yrs, fellowship 16.9 +/-9.5 yrs | Survey or Question-naire | Clinical Practice Education Research |

*(Continued)*

**Table 2.** (Continued)

| Author Year | Country Setting | Methodology Study Design | Aim of Study | PL-EP | Duration (dur) Accredited (acrd) Area of Practice (aop) | Key Features of PL-EP | n age (avg +/- SD) exp (avg +/- SD) | Data Collect Tool | Pillars of Advanced Practice |
|---|---|---|---|---|---|---|---|---|---|
| Camden 2015 [53] | Canada Online | Mixed Methods Pre-Post Design | to evaluate the immediate and short term (two month) impact of an evidence based online Developmental Co-ordination Disorder (DCD) module tailored for PTs on self reported knowledge and skills and evidence based practice | Single Encounter Course | dur: not reported acrd: no aop: pediatric | Didactic Approach Case Study Approach Self Reflection | n: 50 age: not reported exp: not reported | Survey or Questionnaire | Clinical Practice Research |
| Carr 2020 [54] | UK Private Practice | Qualitative Grounded Theory | to explore how MSK PTs exposed to regular observed clinical practice with formal graded feedback, considered this approach to support their development of clinical expertise | Mentorship | dur: not reported acrd: no aop: MSK | Mentorship Student Assessment | n: 11 age: not reported exp: not reported | Semi-Structured Interviews | Clinical Practice Leadership Education |
| Cheema 2022 [55] | USA Online | Quantitative Retrospective Analysis | to examine the relationship between levels of clinician training and patient-reported outcomes in the treatment of CLBP | Residency and Fellowship | dur: not reported acrd: not reported aop: MSK | Mentorship | n: 111 (52 non ed; 69 res or fellow) age: 52% > 40 yrs; 42% > 40 yrs exp: 38% > 21 yrs; 17% > 21 yrs | Patient Outcomes Survey or Questionnaire | Clinical Practice Leadership |
| Chipchase 2016 [56] | Australia Not reported | Quantitative RCT | to determine whether a traditional workshop with a single follow-up meeting with the educator was more likely to change practice behaviour and patient outcomes than a traditional workshop with no opportunity for follow-up | Single Encounter Course | dur: 2 days acrd: no aop: MSK | Didactic Approach Case Study Approach Skill Practice | n: 23 (ctrl 11, int 12) age: ctrl 41.91yrs; int 41.25 yrs exp: ctrl 18 yrs; int 18.08 yrs | Patient Outcomes Survey or Questionnaire | Clinical Practice |
| Cimdi 2012 [57] | Australia Rehabilitation Hospital | Quantitative Survey Methodology | to examine whether participation in an EBP professional development workshop is an effective strategy to enhance knowledge, attitudes and behaviours associated with EBP of PTs working in a rehabilitation setting | Single Encounter Course | dur: 3 hours acrd: no aop: non specific | Didactic Approach Case Study Approach Skill Practice | n: 17 age: not reported exp: 2–27 yrs (range) | Survey or Questionnaire | Education Research |
| Cleland 2009 [58] | USA Private Clinic | Quantitative Pre-Post Design | to investigate the effectiveness of an ongoing educational (OE) intervention for improving the outcomes for patients with neck pain | Multiple Encounter Course | dur: 8hrs over 2 days (+/- 4hrs over 4–7weeks) acrd: no aop: MSK | Didactic Approach Mentorship Skill Practice or Practical Application | n: 19 (OE 10; ctrl 9) age: 41 (OE 38.5; ctrl 43.8) yrs exp: 13.6 (OE 11; ctrl 16.4) yrs | Patient Outcomes System level impact | Clinical Practice Leadership |

*(Continued)*

| Author Year | Country Setting | Method-ology Study Design | Aim of Study | PL-EP | Duration (dur) Accredited (acrd) Area of Practice (aop) | Key Features of PL-EP | n age (avg +/- SD) exp (avg +/- SD) | Data Collect Tool | Pillars of Advanced Practice |
|---|---|---|---|---|---|---|---|---|---|
| Constantine 2012 [59] | UK University | Qualitative Phenomenology | to explore physiotherapists' experiences of change and/or development in their clinical practice after successfully completing a Masters in manual therapy degree | Masters Level Program | dur: 1–2 yrs acrd: yes aop: MSK | Didactic Approach Case Study Approach Mentorship Skill Practice Student Assessment Research assignments | n: 7 age: 32–50 yrs (range) exp: not reported | Semi-Structured Interviews | Clinical Practice Leadership Education Research |
| Cowell 2019 [60] | UK Primary Care Department | Qualitative Phenomenology | to understand the impact of a formal training programme in CFT on 10 PTs, including novices with no prior exposure to the concept | Multiple Encounter Course | dur: 10 months acrd: no aop: non-specific | Didactic Approach Case Study Approach Mentorship Self Reflection Skill Practice Student Assessment | n: 10 age: not reported exp: 6 yrs, 3–14 yrs (range) | Semi-Structured Interviews | Clinical Practice Leadership Education |
| Cunningham 2017 [61] | Kenya Medical Training College | Quantitative Cross sectional study | to determine the influence of a post-graduate, orthopedic residency program on the participant's knowledge, clinical reasoning, and psychomotor skills related to the examination and evaluation of musculoskeletal conditions | Residency and Fellowship | dur: 18 months acrd: yes aop: MSK | Didactic Approach Case Study Approach Mentorship Self Reflection Skill Practice Student Assessment | n: 22 (grad 12, ctrl 10) age: grad 35.4 yrs; ctrl 33.3 yrs exp: grad 11.08 yrs; ctrl 9.4 yrs | Clinical Reasoning Ax Mentor or Instructor Evaluation | Clinical Practice |
| Cunningham 2019 [62] | Kenya | Mixed Methods Pre-Post Design | to explore the clinical reasoning development of physical therapists participating in an 18-month orthopaedic residency program in Nairobi, Kenya | Residency and Fellowship | dur: 18 months acrd: yes aop: MSK | Didactic Approach Mentorship Skill Practice Student Assessment | n: 14 age: 32.3 yrs exp: 9 yrs | Clinical Reasoning Ax | Clinical Practice |

*(Continued)*

| Author Year | Country Setting | Methodology Study Design | Aim of Study | PL-EP | Duration (dur) Accredited (acrd) Area of Practice (aop) | Key Features of PL-EP | n age (avg +/- SD) exp (avg +/- SD) | Data Collect Tool | Pillars of Advanced Practice |
|---|---|---|---|---|---|---|---|---|---|
| Cunningham 2021 [63] | Kenya Individual participant practice settings | Qualitative Phenomenology | to explore the influence of an international partnership developed to provide advanced education to physical therapists in Kenya and the impact of the program on clinical practice | Residency and Fellowship | dur: 18 months acrd: yes aop: MSK | Didactic Approach Mentorship Skill Practice | n: 14 age: 29 yrs (median) exp: 8.2 yrs (median) | Semi-Structured Interviews | Clinical Practice Leadership Education |
| Cunningham 2022 [64] | Kenya Residency Site | Mixed Methods Phenomenology | to explore the influence of residency training on the professional development of PTs | Residency and Fellowship | dur: 18 months acrd: yes aop: MSK | Didactic Approach Case Study Approach Mentorship Self Reflection Skill Practice Student Assessment | n: 27 age: 33.3 yrs exp: 9.7 yrs | Semi-Structured Interviews Survey or Questionnaire | Clinical Practice Leadership Education Research |
| De Rooij 2020 [65] | Netherlands Primary Care | Mixed Methods Pre-Post Design | to evaluate the effect of an educational course on competence (knowledge and clinical reasoning) of primary care physical therapists (PTs) in treating patients with knee osteoarthritis (KOA) and comorbidity according to the developed strategy | Multiple Encounter Course | dur: 6 months acrd: no aop: MSK | Didactic Approach Case Study Approach Skill Practice | n: 34 age: 43.7 ± 11.1 yrs exp: 38% 0–15 yrs, 62% > 15 yrs | Clinical Reasoning Ax Semi-Structured Interviews Survey or Questionnaire | Clinical Practice |
| Demmelmaier 2012 [66] | Sweden Primary Care | Quantitative Longitudinal Observation | to evaluate an educational model by performing a tailored skills training intervention for caregivers and studying changes over time in physiotherapists' assessment of prognostic factors in telephone consultations | Multiple Encounter Course | dur: 15 hours over 20 weeks acrd: no aop: MSK | Didactic Approach Case Study Approach Skill Practice | n: 4 age: 36.5 yrs, 27–44 (range exp: 12.25 yrs, 4–20 (range) | Mentor or Instructor Evaluation | Clinical Practice |
| Dennis 1987 [67] | Australia Private Practice | Quantitative RCT | to compare the clinical behaviour, defined as time allocation between different assessment and treatment procedures, of 16 generalist physiotherapists (Gen) and 16 manipulative therapists (MT) | Accredited Area of Practice Training | dur: not reported acrd: yes aop: MSK | Skill Practice | n: 32 (16 Gen, 16 MT) age: not reported exp: Gen 13 yrs, MT 6 yrs | Observation | Clinical Practice |

*(Continued)*

**Table 2.** (Continued)

| Author Year | Country Setting | Method-ology Study Design | Aim of Study | PL-EP | Duration (dur) Accredited (acrd) Area of Practice (aop) | Key Features of PL-EP | n age (avg +/- SD) exp (avg +/- SD) | Data Collect Tool | Pillars of Advanced Practice |
|---|---|---|---|---|---|---|---|---|---|
| Deutscher 2014 [68] | Israel Out-patient Clinic | Quantitative Longitudinal Observation | to examine associations between McKenzie training, functional status (FS) at discharge, and number of physical therapy visits (utilization) in patients receiving physical therapy for low back pain | Single Encounter Course | dur: 28 hrs acrd: yes aop: MSK | Didactic Approach Case Study Avpproach Skill Practice Student Assessment | n: 195 age: 42 +/- 9 yrs [28–65 range] exp: 13 +/- 7 yrs [7–46 range] | Patient Outcomes | Clinical Practice |
| Dizon 2014 [69] | Phillipines Not Reported | Quantitative RCT | to assess the effectiveness of the contextually-based EBP training program in improving knowledge, skills, attitudes and behaviour of Filipino physical therapists | Single Encounter Course | dur: 1 day acrd: no aop: non-specific | Didactic Approach Skill Practice | n: 54 (Int 27 Ctrl 27) age: Int 29 Ctrl 28 yrs (median) exp: Int 4.2 Ctrl 3 yrs (median) | Clinical Reasoning Ax Survey or Question-naire | Research |
| Fary 2015 [70] | Australia Public and private settings | Quantitative RCT | 1) to evaluate the effectiveness of the RAP-eL resource in achieving increased self-reported confidence and knowledge in managing people with RA among Australian physiotherapists 2) to evaluate the retention of that confidence and knowledge over the short-term | Multiple Encounter Course | dur: 4 modules over 4 weeks acrd: No aop: MSK (RA) | Didactic Approach Case Study Approach | n: 104 (Ed 56; ctrl 48) age: not reported exp: ed 13.45 yrs, ctrl 16.06 yrs | Survey or Question-naire | Clinical Practice |
| Furze 2019 [71] | USA | Qualitative Retrospective Analysis | to use narrative as a teaching and learning tool to gain insight into the progressive development of the residents' learning process | Residency and Fellow-ship | dur: 1 yr acrd: yes aop: MSK and Pediatric | Didactic Approach Mentor-ship Self Reflection Skill Practice Student Assessment | n: 6 age: not reported exp: 0–2.5 yrs (range) | Content Analysis of Written Narratives | Clinical Practice Leadership |

*(Continued)*

**Table 2.** (Continued)

| Author Year | Country Setting | Methodology Study Design | Aim of Study | PL-EP | Duration (dur) Accredited (acrd) Area of Practice (aop) | Key Features of PL-EP | n age (avg +/- SD) exp (avg +/- SD) | Data Collect Tool | Pillars of Advanced Practice |
|---|---|---|---|---|---|---|---|---|---|
| Green 2008 [72] | UK University | Mixed Methods Retrospective Analysis | to explore the career pathways of these graduates and the influence of Master's education on their careers. With new ways of working within the NHS and opportunities for consultant, extended and specialist roles, the study set out to establish where these graduates are now working and in what roles, and how Masters education had influenced these positions. | Masters Level Program | dur: 1 yr acrd: yes aop: MSK | Didactic Approach Mentorship Skill Practice Student Assessment | n: 48 age: not reported exp: not reported | Survey or Questionnaire Group Interviews | Clinical Practice Leadership Education Research |
| Hansell 2023 [73] | Australia Intensive care setting | Quantitative Survey Methodology | to evaluate the impact of attending a physiotherapy Lung Ultrasound training course on the acquisition of competence and confidence for physiotherapists in Australia and to determine the barriers and facilitators in achieving such competence. | Single Encounter Course | dur: 10.2 hrs ± 5.3 acrd: yes aop: Cardiorespiratory | Didactic Approach Skill Practice Student Assessment | n: 39 age: 41 +/-9.74 yrs, 28–64 (range) exp: 17.64 +/- 9.5, 7–44 (range) | Survey or Questionnaire | Clinical Practice |
| Harrison 2022 [74] | Australia Public Hospitals | Quantitative Pre-Post Design | 1) to quantify knowledge and skills in evidence-based practice 2) to quantify the barriers to the application of research in the clinical setting 3) to determine the impact of a flipped classroom training program that addresses the core competencies for the teaching of evidence-based practice in registered physical therapists | Multiple Encounter Course | dur: 8 hrs, over 3 months accd: no aop: non-specific | Didactic Approach Case Study Approach Skill Practice | n: 94 age: 74% < 30 yrs exp: 60% < 5 yrs | Survey or Questionnaire | Research |

*(Continued)*

| Author Year | Country Setting | Method-ology Study Design | Aim of Study | PL-EP | Duration (dur) Accredited (acrd) Area of Practice (aop) | Key Features of PL-EP | n age (avg+/- SD) exp (avg+/- SD) | Data Collect Tool | Pillars of Advanced Practice |
|---|---|---|---|---|---|---|---|---|---|
| Heneghan 2022 [75] | UK Online | Mixed Methods Longi-tudinal Observa-tion | 1) to explore the influ-ence of telehealth e-mentoring on health out comes in patients with MSK complaints 2) to explore the develop-ment of critical thinking, clinical reasoning, communication skills and confidence of postgradu-ate mentees engaged in telehealth e-mentoring. 3) to explore the mentor acceptability and appro-priateness of tele health e-mentoring to facilitate student develop-ment towards achievement of IFOMPT Educational Standards | Mentor-ship | dur: 150 hours acrd: Yes aop: MSK | Case Study Approach Mentor-ship Self Reflection Skill Practice Student Assess-ment | n: 10 age: 28 yrs exp: 1.5–20 yrs, (range) | Patient Outcomes; Mentor or Instructor Evaluation Semi-Structured Interviews | Clinical Practice Leadership Education Research |
| Jones 2008 [76] | USA | Quan-titative Survey Methodol-ogy | to describe and compare the professional devel-opment and leadership activities between 2 groups of orthopedic physical therapists: resi-dency and non-residency trained. | Resi-dency and Fellow-ship | dur: not reported acrd: yes aop: MSK | Didactic Approach Mentor-ship Skill Practice | n: res 41, non res 20 age: res 32.5 yrs, non-res 30 yrs (median) exp: not reported | Survey or Question-naire | Education |
| Kafri 2023 [77] | Israel Aca-demic and Work-place | Mixed Methods Pre-Post Design | to evaluate the influ-ence of a "Knowledge Translation-Motor Learning" intervention on ML-related self-efficacy, reported ML imple-mentation, and general perceptions and work environment among certified PTs | Single Encoun-ter Course | dur: 20 hours acrd: no aop: non-specific | Didactic Approach Case Study Approach Skill Practice | n: pre-post 111; f/u 25 age: pre-post 35.0±7.5 yrs, 25–59 (range) f/u 37.5±5.8 yrs, 29–42 (range) exp: pre-post 8.5±7.8 yrs, 0–33 (range); f/u 11±6.7 yrs, 2–30 (range) | Survey or Question-naire | Clinical Practice |
| Karas 2016 [78] | USA Out-patient Centre | Quantita-tive Pre-Post Design | to evaluate the effects of a structured knowledge translation programme on the frequency of man-ual therapy techniques performed by physical therapists on patients with neck pain | Single Encoun-ter Course | dur: not reported acrd: no aop: MSK | Didactic Approach Skill Practice | n: 16 age: not reported exp: 2–30 yrs (range) | Survey or Question-naire | Clinical Practice |

*(Continued)*

**Table 2.** (Continued)

| Author Year | Country Setting | Methodology Study Design | Aim of Study | PL-EP | Duration (dur) Accredited (acrd) Area of Practice (aop) | Key Features of PL-EP | n age (avg +/- SD) exp (avg +/- SD) | Data Collect Tool | Pillars of Advanced Practice |
|---|---|---|---|---|---|---|---|---|---|
| Karvonen 2015 [79] | Finland Healthcare Centre | Quantitative Pre-Post Design | to evaluate how well physical therapists can learn the skills associated with pathoanatomical classification of patients with LBP consequent to training provided in a continuing education short course | Multiple Encounter Course | dur: 5 days acrd: no aop: MSK | Didactic Approach; Mentorship Skill Practice | n: 6 age: 32 yrs exp: 12 yrs, 4–24 (range) | Mentor or Instructor Evaluation | Clinical Practice |
| Kerssens 1999 [80] | Netherlands Private Practice | Quantitative Pre-Post Design | to study the effectiveness of a training program for the enhancement of patient education skills in physical therapy | Multiple Encounter Course | dur: 28 hrs over 6 months acrd: no aop: non-specific | Didactic Approach Case Study Approach | n: 19 age: not reported exp: not reported | Survey or Questionnaire | Clinical Practice |
| Lambrinos 2023 [81] | Australia Online | Quantitative RCT | to evaluate if an online education course could improve physiotherapists' confidence and competence in the prescription and application of Mechanical Insufflation-Exsufflation compared with no education | Single Encounter Course | dur: 6 hrs acrd: no aop: Cardiorespiratory | Didactic Approach Case Study Approach Student Assessment | n: 66 (ctrl 38, int 28) age: not reported exp: ctrl 5.5 +/- 5.9 yrs, int 6.5 +/- 7.4 yrs | Survey or Questionnaire | Clinical Practice |
| Lane 2022 [82] | USA Outpatient Clinics | Quantitative RCT | to determine the effectiveness of providing PTs with Pain Neuroscience Education (PNE) training on patient-centered outcomes for patients with chronic neck or back pain receiving PT under routine clinical circumstances | Single Encounter Course | dur: 16 hours acrd: no aop: pain | Didactic Approach Skill Practice | n: 115 age: 30.8 +/-6.2 yrs exp: 4.7 +/-5.9 yrs | Patient Outcomes Survey or Questionnaire | Clinical Practice |
| Lawford 2018 [83] | Australia Private Health Settings | Qualitative Case Study | to explore physical therapists' experiences with, and the impacts of, a training program in person-centered practice to support exercise adherence in people with knee osteoarthritis | Multiple Encounter Course | dur: 3-months (3 training days total) acrd: no aop: MSK (OA) | Didactic Approach Case Study Approach Skill Practice | n: 8 age: 35 +/- 8 yrs exp: 14 +/- 8 yrs | Semi-Structured Interviews | Clinical Practice |
| Lawford 2019 [84] | Australia Private Practice | Quantitative Case Study | to audit consultations to determine how well physiotherapists implemented person-centred practice principles and behaviour change techniques into patient consultations after participation in a 2-day training work-shop | Single Encounter Course | dur: 2 days (16 hrs) acrd: no aop: non-specific | Didactic Approach Self Reflection Skill Practice | n: 8 age: 35 yrs, 26–50 (range) exp: 14 yrs, 4–28 (range | Mentor or Instructor Evaluation | Clinical Practice |

*(Continued)*

**Table 2.** (Continued)

| Author Year | Country Setting | Method-ology Study Design | Aim of Study | PL-EP | Duration (dur) Accredited (acrd) Area of Practice (aop) | Key Fea-tures of PL-EP | n age (avg +/- SD) exp (avg +/- SD) | Data Collect Tool | Pillars of Advanced Practice |
|---|---|---|---|---|---|---|---|---|---|
| Levsen 2001 [85] | USA Private Practice | Quantita-tive Pre-Post Design | to investigate whether a long-term course emphasizing clinical rea-soning enhances clinical outcomes | Multiple Encoun-ter Course | dur: 1 year acrd: no aop: MSK | Didactic Approach Case Study Approach Skill Practice | n: 6 (3 ed, 3 non ed) age: not reported exp: minimum 5 yrs | Patient Outcomes | Clinical Practice |
| Lonsdale 2017 [86] | Ireland Out-patient clinics | Quantita-tive RCT | to assess the effect of an intervention designed to enhance physiother-apists' communication skills on patients' adher-ence to recommenda-tions regarding home-based rehabilitation for chronic low back pain | Single Encoun-ter Course | dur: 8 hrs acrd: no aop MSK | Didactic Approach Case Study Approach Skill Practice | n: 50 age: 32.24 +/- 5.26 yrs exp: 9.9 +/- 5.16 yrs | Patient Outcomes | Clinical Practice |
| Louw 2022 [87] | USA Online | Mixed Methods Pre-Post Design | to determine if therapists attending a self-paced 3-hour online Pain Neuroscience Education (PNE) program was associated with any observed changes to patient outcomes and also clinical practice | Single Encoun-ter Course | dur: 3 hrs acrd: no aop: pain | Didactic Approach | n: 25 age: not reported exp: 9.2 +/- 9 yrs, 2–29 (range) | Patient Outcomes System level impact | Clinical Practice |
| Maas 2012 [88] | Nether-lands Private Practice and Hospital | Quantita-tive RCT | to compare the effective-ness of peer assessment (PA) with the usual case discussion (CD) strategy on adherence to CPGs for physical therapist management of upper extremity complaints | Multiple Encoun-ter Course | dur: 4 sessions over 6 months acrd: no aop: upper extremity | Case Study Approach Self Reflection Skill Practice | n: 149 (PA 73, CD 76) age: PA 45.15 +/- 11.03 yrs; CD 44.76 +/- 9.74 yrs exp: PA 20.42 +/- 11.37 yrs; CD 20.86 +/- 9.71 yrs | Mentor or Instructor Evaluation Survey or Question-naire | Clinical Practice Education |
| MacPher-son 2019 [89] | USA | Qual-itative Phenom-enologoy | to elucidate graduate perceptions of how fel-lowship training impacted their post-training profes-sional and personal lives | Resi-dency and Fellow-ship | dur: not reported acrd: yes aop: MSK | Didactic Approach Mentor-ship Self Reflection Skill Practice Student Assess-ment | n: 13 age: 33–53 yrs (range) exp: 7.5–31 yrs (range) | Semi-Structured Interviews | Clinical Practice Leadership Education |

*(Continued)*

**Table 2.** (Continued)

| Author Year | Country Setting | Method-ology Study Design | Aim of Study | PL-EP | Duration (dur) Accredited (acrd) Area of Practice (aop) | Key Features of PL-EP | n age (avg +/- SD) exp (avg +/- SD) | Data Collect Tool | Pillars of Advanced Practice |
|---|---|---|---|---|---|---|---|---|---|
| Madi 2018 [90] | UK University | Mixed Methods Longitudinal Observation | to capture the advancement of clinical reasoning skills throughout and after participating in an MACP approved MSK PT program | Masters Level Program | dur: 1 yr acrd: yes aop: MSK | Didactic Approach Mentorship Self Reflection Skill Practice Student Assessment Research assignments | n: 6 age: 23–34 yrs (range) exp: 2–10 yrs (range) | Clinical Reasoning Ax Semi-Structured Interviews Survey or Questionnaire | Clinical Practice Leadership Education Research |
| Mansell 2020 [91] | UK Acute Hospital Trust | Mixed Methods Pre-Post Design | to evaluate the addition of SBE to an on-call training programme on non-respiratory physiotherapists' self-evaluated confidence. Additionally, the study aimed to evaluate if SBE facilitates identification of learning needs. | Multiple Encounter Course | dur: 1 year acrd: no aop: Cardiorespiratory | Didactic Approach Case Study Approach Skill Practice | n: 10 age: not reported exp: 1.5–7 yrs (range) | Semi-Structured Interviews Survey or Questionnaire | Clinical Practice Leadership Education |
| March 2024 [92] | Australia Medical Simulation Centre | Mixed Methods Pre-Post Design | to develop and evaluate a simulation-based educational strategy for musculoskeletal physiotherapists to improve knowledge and confidence in patient-centred care | Single Encounter Course | dur: 3 hrs acrd: no aop: MSK | Didactic Approach Case Study Approach Skill Practice Simulation Based Training | n: 22 age: 20–30 yrs: 36%, 31–40 yrs: 36%, 41–50 yrs: 14%, 51+yrs: 14% exp: not reported | Survey or Questionnaire | Clinical Practice |
| Murray 2015 [93] | Ireland Hospital Out-patient Clinics | Quantitative RCT | to examine the effects of communication skills training on physiotherapists' supportive behaviour during clinical practice. | Single Encounter Course | dur: 8 hrs acrd: no aop: MSK | Didactic Approach Case Study Approach Skill Practice | n: 24 (12 ed, 12 ctrl) age: ed 32.67 +/- 3.28 yrs; ctrl 34.92 +/- 5.98 yrs exp: ed 8.83 +/- 3.67 yrs; ctrl 10.17 +/- 5.03 yrs | Mentor or Instructor Evaluation Survey or Questionnaire | Clinical Practice |
| Naidoo 2022 [94] | USA | Qualitative Retrospective Analysis | to investigate (through thematic analysis of reflective narratives) whether residents who were exposed to a specific clinical reasoning strategy showed development in the type and number of reasoning strategies used over the course of residency training. | Residency and Fellowship | dur: 1 year acrd: yes aop: MSK | Didactic Approach Case Study Approach Mentorship Self Reflection Skill Practice Student Assessment | n: 5 age: 25.8 yrs, 24–29 (range) exp: 0–5 yrs | Clinical Reasoning Ax Content Analysis of Narratives | Clinical Practice Education |

*(Continued)*

**Table 2.** (Continued)

| Author Year | Country Setting | Method-ology Study Design | Aim of Study | PL-EP | Duration (dur) Accredited (acrd) Area of Practice (aop) | Key Features of PL-EP | n age (avg +/- SD) exp (avg +/- SD) | Data Collect Tool | Pillars of Advanced Practice |
|---|---|---|---|---|---|---|---|---|---|
| Ntou-menopou-los 2017 [95] | Australia Acute Care | Quan-titative Pre-Post Design | to evaluate the impact of a one-day DTU curric-ulum course (including pre-reading, didactic lectures, practical training and image recognition skills) on the short-term knowledge acquisition by group of acute care physiotherapists | Single Encoun-ter Course | dur: 1 day acrd: no aop: Cardiore-spiratory | Didactic Approach Skill Practice | n: 12 age: not reported exp: not reported | Survey or Question-naire | Clinical Practice |
| Olsen 2015 [96] | Norway Hospitals | Quan-titative Pre-Post Design | to assess the short and long term impact of a six-month multifaceted and clinically integrated training program in EBP on the knowledge, skills, beliefs and behaviour of CIs supervising physio-therapy students. | Multiple Encoun-ter Course | dur: 6 months acrd: no aop: non-specific | Didactic Approach Mentor-ship Self Reflection Skill Practice Student Assess-ment | n: 29 (int 14, ctrl 15) age: int 40.5 yrs, 26–55 (range) ctrl 38.9 yrs, 26–61 (range) exp: int 13.6 yrs, 2–28 (range) ctrl 12.2 yrs, 2–32 (range) | Survey or Question-naire | Clinical Practice Research |
| Overmeer 2009 [97] | Sweden Private Clinic and Health Centres | Quantita-tive Pre-Post Design | to examine the effects of an 8-day university-based training course, aimed at identifying and addressing psychoso-cial prognostic factors during phys- iotherapy treatment, in shifting therapists towards a more biopsychosocial orientation as measured by changes in beliefs/atti-tudes, knowledge, skills and behaviour. | Multiple Encoun-ter Course | dur: 8 days over 8 weeks acrd: yes aop: MSK | Didactic Approach Case Study Approach Skill Practice | n: 42 age: 45.8 +/-6.8 yrs exp: 18.5 +/-7.9 yrs | Mentor or Instructor Evaluation Survey or Question-naire | Clinical Practice Research |
| Overmeer 2011 [98] | Sweeden Primary Care Practice | Quantita-tive RCT | to examine the effects on outcomes (pain and disability) in patients of a course about psycho-social prognostic factors for PTs | Multiple Encoun-ter Course | dur: 8 days acrd: yes aop: MSK | Didactic Approach Case Study Approach Skill Practice | n: 42 (ed: 22; ctrl: 20) age: 45.8 (ed 47.1, ctrl 43.7) yrs exp: 18.5 (ed18.4, ctrl 15.8) yrs | Patient Outcomes Survey or Question-naire | Clinical Practice |
| Perry 2011 [99] | UK University | Qual-itative Phenom-enologoy | to explore the profes-sional and personal impact that a clinical Masters program of manipulative therapy edu-cation had on the lives of individuals who had undertaken the course and was a follow-on study of participants' career pathways following Mas-ters education | Masters Level Program | dur: 1 yr acrd: yes aop: MSK | Didactic Approach Mentor-ship Skill Practice Student Assess-ment | n: 7 age: 38–42 yrs (range) exp: 21–26 yrs (range) | Focus Groups | Clinical Practice Leadership Education Research |

*(Continued)*

**Table 2.** (Continued)

| Author Year | Country Setting | Methodology Study Design | Aim of Study | PL-EP | Duration (dur) Accredited (acrd) Area of Practice (aop) | Key Features of PL-EP | n age (avg +/- SD) exp (avg +/- SD) | Data Collect Tool | Pillars of Advanced Practice |
|---|---|---|---|---|---|---|---|---|---|
| Peter 2013 [100] | Netherlands Primary Care and Rehab Centres | Quantitative RCT | to develop and compare two educational courses, i.e., an interactive course (IW) and a conventional presentation (CE), with respect to their ability to improve satisfaction, knowledge and guideline adherence. | Single Encounter Course | dur: IW 3 hr; CE 2 hrs acrd: yes aop: MSK (OA) | Didactic Approach Case Study Approach Skill Practice | n: IW 108, CE 95 age: IW 43.9 +/-11.1 yrs, CE 42.8 +/-12.8 yrs exp: IW 74% > 10 yrs, CE 66% > 10 yrs | Survey or Questionnaire | Clinical Practice |
| Peter 2015 [101] | Netherlands Primary Care and Rehab Centres | Quantitative RCT | to determine, on the national level, the effectiveness of an interactive, postgraduate educational intervention. | Single Encounter Course | dur: 3 hours acrd: yes aop: MSK (OA) | Case Study Approach Skill Practice | n: int 133 ctrl 151 age: int 45.7 +/-10.6 yrs, ctrl 45.4 +/- 11.9 yrs exp: int 79.7% > 10 yrs, ctrl 72.9% > 10 yrs | Survey or Questionnaire | Clinical Practice |
| Petty 2011(a) [102] | UK Private Practice and Hospitals | Qualitative Naturalistic Inquiry | to explore the impact of an MACP approved MSc on practitioners and offers a conceptual model of their development towards clinical expertise. | Masters Level Program | dur: 1 year acrd: yes aop: MSK | Didactic Approach Mentorship Case Study Approach Self Reflection Skill Practice Student Assessment | n: 11 age: 38.2 yrs, 31–52 (range) exp: 8.5 yrs, 3–24 (range) | Semi-Structured Interviews | Clinical Practice Leadership Education Research |
| Petty 2011(b) [103] | UK Private Practice and Hospital | Qualitative Grounded Theory | to develop an explanatory theory of the learning transition of neuromusculoskeletal physiotherapists on completion of an MACP-approved MSc programme. | Masters Level Program | dur: 1 year acrd: yes aop: MSK | Didactic Approach Mentorship Case Study Approach Self Reflection Skill Practice Student Assessment | n: 11 age: 38.2 yrs, 31–52 (range) exp: 8.5 yrs, 3–24 (range) | Semi-Structured Interviews | Clinical Practice Leadership Education Research |

*(Continued)*

**Table 2.** (Continued)

| Author Year | Country Setting | Methodology Study Design | Aim of Study | PL-EP | Duration (dur) Accredited (acrd) Area of Practice (aop) | Key Features of PL-EP | n age (avg +/- SD) exp (avg +/- SD) | Data Collect Tool | Pillars of Advanced Practice |
|---|---|---|---|---|---|---|---|---|---|
| Prizinski 2021 [104] | USA | Qualitative Participatory Action Research | to investigate and develop a coaching program to address the developmental needs of Novice Physical Therapists (NPTs) within residency program. This was approached by understanding residents and mentors expectations, ways to enhance mentor-resident interactions, and develop clinical reasoning skills for NPTs learning within a community of practice (CoP) | Residency and Fellowship | dur: 1 yr acrd: yes aop: MSK | Mentorship Student Assessment | n: 8 age: 26.5 yrs exp: <2 yrs | Clinical Reasoning Ax Mentor or Instructor Evaluation Semi-Structured Interviews | Clinical Practice Leadership |
| Rebbeck 2006 [105] | Australia Private Practice | Quantitative RCT | to evaluate the effect of an active implementation strategy that included education by opinion leaders compared with a passive implementation strategy that consisted of dissemination of the guidelines only. | Single Encounter Course | dur: 8 hrs acrd: no aop: MSK | Didactic Approach Case Study Approach Skill Practice | n: 27 (int 14, ctrl 13) age: not reported exp: not reported | Patient Outcomes Survey or Questionnaire System level impact | Clinical Practice |
| Resnik 2004 [106] | USA Outpatient Rehab Facilities | Quantitative Cross sectional study | to assess outcomes of care, as measured by changes in patient self-report of health status, for patients with lumbar impairments treated by clinicians 1) who had an orthopaedic clinical specialist certification (OCS), 2) who had graduated from residency programs approved by American Academy of Orthopaedic Manual Therapy (AAOMPT), or 3) who had obtained miscellaneous certification in manual therapy (MTC). | Accredited Area of Practice Training | dur: not reported acrd: yes aop: MSK | Mentorship | n: 930 age: not reported exp: 10.8 yrs | Patient Outcomes | Clinical Practice |

*(Continued)*

| Author Year | Country Setting | Method-ology Study Design | Aim of Study | PL-EP | Duration (dur) Accredited (acrd) Area of Practice (aop) | Key Fea-tures of PL-EP | n age (avg +/- SD) exp (avg +/- SD) | Data Collect Tool | Pillars of Advanced Practice |
|---|---|---|---|---|---|---|---|---|---|
| Rode-ghero 2015 [107] | USA Out-patient rehabil-itation settings | Quantita-tive Ret-rospective Analysis | to investigate the impact of completing an accredited residency or fellowship program on clinical outcomes in patients with MSK conditions. A secondary aim was to provide initial insight about the value of advanced post-professional education for the PT profession. | Resi-dency and Fellow-ship | dur: not reported acrd: yes aop: MSK | Mentor-ship | n: 363 (no ed 306, res 45, fel 12) age: no ed 43% 24–35 yrs, 57% 36–45 + yrs res 62% 24–35yrs, 38% 36–45 + yrs fel 25% 24–35 yrs, 75% 36–45 + yrs exp: no ed 41% 0–9 yrs, 59% 10 + yrs res 67% 0–9 yrs, 33% 10 + yrs fel 17% 0–9 yrs, 83% 10 + yrs | Patient Outcomes Survey or Question-naire | Clinical Practice Leadership |
| Rushton 2010 [108] | UK | Qual-itative Phenom-enologoy | to identify behaviours indicative of masters level (post-graduate) practice in manipulative physiotherapy to inform ongoing development of educational standards. | Masters Level Program | dur: 1 yr acrd: yes aop: MSK | Didactic Approach Mentor-ship Case Study Approach Self Reflection Skill Practice Student Assess-ment | n: 13 age: not reported exp: 6.3 yrs, 4–9 (range) | Clinical Reasoning Ax Mentor or Instructor Evaluation; Semi-Structured Interviews | Clinical Practice Leadership Education Research |
| Schreiber 2012 [109] | USA school-based settings | Quantita-tive Pre-Post Design | to investigate the effect of knowledge translation procedures as part of a continuing education conference for pediatric physical therapists on knowledge and frequency of use of tests and measures. | Multiple Encoun-ter Course | dur: 90 mins + 16 weeks acrd: no aop: Pediatrics | Didactic Approach Case Study Approach Self Reflection | n: 8 age: not reported exp: > 1–30 yrs (range) | Survey or Question-naire | Clinical Practice |
| Seif 2019 [110] | USA Private Practice | Quantita-tive Pre-Post Design | to determine whether a course series that included review sessions, between-course assign-ments, and a practical and written examination changed clinician 1) atti-tudes (i.e., comfort and confidence in working with patients with spinal dysfunctions) and 2) behaviors (i.e., utilization of outcome measures). | Multiple Encoun-ter Course | dur: not reported acrd: no aop: MSK | Didactic Approach Case Study Approach Skill Practice Student Assess-ment | n: 24 age: not reported exp: 1–20 + yrs (range) | Survey or Question-naire | Clinical Practice |

*(Continued)*

| Author Year | Country Setting | Method-ology Study Design | Aim of Study | PL-EP | Duration (dur) Accredited (acrd) Area of Practice (aop) | Key Features of PL-EP | n age (avg+/- SD) exp (avg+/- SD) | Data Collect Tool | Pillars of Advanced Practice |
|---|---|---|---|---|---|---|---|---|---|
| Shalabi 2024 [111] | Saudi Arabia. Rehabilitation Sciences Department | Quantitative Cross sectional study | to assess PTs' satisfaction with and attitudes towards Online Continuing Medical Education (OCME) and its impact on their clinical practice. And to examine the factors that affect the findings for satisfaction, attitude, and impact towards OCME. | Single Encounter Course | dur: not reported acrd: no aop: non specific | Not Specified | n: 127 age: 25–29 yrs 56; 30–39 yrs 40;>40 yrs 31 exp: 0–5 yrs 55; 6–15 yrs 43;>15 yrs 29 | Survey or Questionnaire | Clinical Practice |
| Sjodahl-Hammarlund 2013 [112] | Sweden | Qualitative Phenomenology | to describe how practicing physiotherapists experienced their learning during participation in inquiry-based online courses. | Masters Level Program | dur: 10 weeks, 200 hrs acrd: yes aop: non specific | Self Reflection Skill Practice Student Assessment Research assignments | n: 5 age: 41 yrs exp: not reported | Survey or Questionnaire | Clinical Practice Leadership Education Research |
| Smith 1999 [113] | USA | Quantitative Cross sectional study | to establish a profile of the typical graduate and to determine the value and influence of residency training on professional development, particularly on clinical expertise | Residency and Fellowship | dur: 1 yr acrd: yes aop: MSK | Mentorship Skill Practice Student Assessment | n: 90 age: 30.4 +/- 0.9 yrs exp: 6.3 +/- 0.4 yrs | Survey or Questionnaire | Clinical Practice Leadership Education Research |
| Souter 2019 [114] | USA | Quantitative Cross sectional study | to examine the influence of education, training, and experience on PTs' diagnostic reasoning in musculoskeletal patient cases | Residency and Fellowship | dur: variable acrd: yes aop: variable | Mentorship Self Reflection Skill Practice Student Assessment | n: 292 age: not reported exp:< 1yr 15, 1–5yrs 93, 6–10 yrs 55, 11–15 yrs 51, 16–20 yrs 32,>20yrs 70 | Clinical Reasoning Ax Survey or Questionnaire | Clinical Practice |
| Stathopoulos 2003 [115] | UK | Qualitative Phenomenology | to explore the impact of undertaking study at master's level on practising physiotherapists from the perspectives of the master's qualified physiotherapists themselves. | Masters Level Program | dur: 1 yr acrd: yes aop: MSK | Didactic Approach Skill Practice Rearch assignments | n: 5 age: 31–40 yrs (range) exp: 6–16 yrs (range) | Semi-Structured Interviews Focus Groups | Clinical Practice Leadership Education Research |
| Stevenson 2006 [116] | UK Hospital | Quantitative RCT | to investigate whether an evidence-based educational package based on the bio-psychosocial management of low back pain would lead to a measurable change in their clinical practice. | Single Encounter Course | dur: 5 hrs acrd: no aop: MSK | Didactic Approach | n: 30 (17 Int, 13 ctrl) age: not reported exp: not reported | Patient Outcomes Survey or Questionnaire | Clinical Practice |

*(Continued)*

| Author Year | Country Setting | Method-ology Study Design | Aim of Study | PL-EP | Duration (dur) Accredited (acrd) Area of Practice (aop) | Key Features of PL-EP | n age (avg +/- SD) exp (avg +/- SD) | Data Collect Tool | Pillars of Advanced Practice |
|---|---|---|---|---|---|---|---|---|---|
| Stevenson 2020 [117] | UK Hospital (NHS) | Qualitative Cross sectional study | to describe the development, implementation and evaluation of a 'Musculoskeletal Interface Service Clinical Trainee Development Programme' that supported three Physiotherapists in their development from a Band 7 to an APP (Band 8a) over a 12-month period. | Residency and Fellowship | dur: 1 yr acrd: no aop: MSK | Didactic Approach; Case Study Approach Mentorship Self Reflection Skill Practice Student Assessment | n: 3 age: 30–34 yrs (range) exp: 9–13 yrs (range) | Mentor or Instructor Evaluation Content Analysis of Narrative | Clinical Practice Leadership Education Research |
| Swinkels 2015 [118] | Netherlands Private Practice and Nursing Homes | Quantitative Pre-Post Design | to describe the development of an educational programme for physiotherapists in the Netherlands, two toolkits of measurement instruments, and the evaluation of an implementation strategy | Multiple Encounter Course | dur: 4 half days over 4–5 months acrd: No aop: non-specific | Didactic Approach Case Study Approach Skill Practice | intervention group (I) n:234 control group (C) n: 199 age: I < 30yrs 35, 30–50 yrs 98, > 50yrs 101 C < 30yrs 47, 30–50yrs 107, > 50yrs 46 exp: I < 10yrs 42, 11–20yrs 40, 21–30yrs 88, > 30yrs 64 C < 10yrs 64, 11–20yrs 38, 21–30yrs 60, > 30yrs 37 | Survey or Questionnaire | Clinical Practice |
| Synnott 2016 [119] | Ireland Private and Public Sector | Qualitative Phenomenology | to understand what physiotherapists' perspectives on treating the biopsychosocial dimensions of chronic low back pain after receiving intensive biopsychosocial training. | Multiple Encounter Course | dur: 9 x 12 hrs acrd: no aop: MSK | Didactic Approach Case Study Approach Mentorship Skill Practice | n: 13 age: not reported exp: 13 yrs, 5–19 (range) | Semi-Structured Interviews | Clinical Practice Leadership |
| Tilson 2014 [120] | USA Outpatient and Inpatient Settings | Mixed Methods Pre-Post Design | to assess the feasibility of the Physiotherapist driven Education for Actionable Knowledge (PEAK) program with respect to practical implementation, participant reaction, and potential for association with change in participants' evidence based practice attitudes, self-efficacy, knowledge and skills, and self-reported behavior. | Multiple Encounter Course | dur: 6 months acrd: no aop: MSK | Case Study Skill Practice | n: 18 age: 34.7 yrs; 27–51 (range) exp: 7.7 yrs; 2–20 (range) | Semi-Structured Interviews Survey or Questionnaire | Clinical Practice Research |

*(Continued)*

**Table 2.** (Continued)

| Author Year | Country Setting | Method-ology Study Design | Aim of Study | PL-EP | Duration (dur) Accredited (acrd) Area of Practice (aop) | Key Features of PL-EP | n age (avg +/- SD) exp (avg +/- SD) | Data Collect Tool | Pillars of Advanced Practice |
|---|---|---|---|---|---|---|---|---|---|
| Westervelt 2020 [121] | USA Outpatient and Home-care | Mixed Methods Pre-Post Design | to examine the effects of an online model of clinical mentoring on PTs experiencing professional isolation in an outpatient MSK setting | Mentor-ship | dur: 3 x 1hr over 5 weeks acrd: no aop: MSK | Case Study Approach Mentor-ship | n: 8 age: 33.13 +/- 9.25 yrs exp: 3.71 +/- 5.06 yrs | Survey or Question-naire | Clinical Practice Leadership Education Research |
| Whitman 2020 [122] | USA | Mixed Methods Survey Methodol-ogy | to describe multiple types of educational outcomes, including graduate professional, educational, and research involve-ment; perceptions of the impact of fellowship training (FT) on clinical and professional attri-butes; access to FT; and changes in employment and graduates' annual income | Resi-dency and Fellow-ship | dur: 32.1 months(avg) acrd: yes aop: MSK | Didactic Approach Case Study Approach Mentor-ship Self Reflection Skill Practice Student Assess-ment | n: 75 age: 39.9 +/-7.8 yrs exp:14.0 +/-8.0 yrs | Survey or Question-naire | Clinical Practice Leadership Education Research |
| Williams 2019 [123] | UK Out-patient Hospital | Quantita-tive RCT | to assess the effective-ness of a work-based mentoring programme to facilitate physiotherapist clinical reasoning on patient outcomes | Mentor-ship | dur: 150 hrs acrd: yes aop: MSK | Mentor-ship Self Reflection Skill Practice Student Assess-ment | n: 16 age: not reported exp: not reported | Clinical Reasoning Ax Patient Outcomes Mentor or Instructor Evaluation | Clinical Practice Leadership |

**Mentorship.** 4 studies investigated mentorship [54,75,121,123]. This PL-EP varies in its duration from as little as 3–150 hours [75,121,123]. It consists of one-on-one as well as small group sessions and is delivered in both online as well as clinical environments. It often includes case studies, group discussion, and observed clinical practice with a focus on clinical reasoning development. It is often focused to the MSK area of practice. Distinct to this pathway is its highly individualized nature, it is frequently self-directed, with ongoing and real time feedback for the PT [54,75,121,123].

**Multiple Encounter Courses.** 22 studies investigated multiple encounter courses [49,51,58,60,87,66,70,74,79,80,83,85,88,91,96–98,109,110,118–120]. This PL-EP is highly variable in its duration, spanning as little as 8 days up to 1 year [85,98]. It is often narrowly focused to the adoption of one specific skill or theory, though it spans a variety of areas of practice [49,60,74,80,91,109,118]. It is consistently delivered via didactic methods and often includes both skill practice and case study discussions and occasionally includes clinical mentorship. Unlike the previously described pathways, education in this form is rarely accredited, and rarely includes formal student assessment [60,96–98,110]. Distinct to this pathway, compared to a single encounter course is the opportunity and encouragement to practice and review between education sessions as well as built in avenues for feedback throughout [49,51,58,60,65,66,70,74,79,80,83,85,88,91,96–98,109,110,118–120].

**Single Encounter Courses.** 26 studies investigated single encounter courses [43,44,46,48,50,53,56,57,68,69,73,77,78,81,82,84,86,87,92,93,95,100,101,105,111,116]. This PL-EP can span anywhere from 3 hours, up to 2 days [46,48].

**Table 3. Quality assessment of included studies (QUADs).**

| Author Year | 1 | 2 | 3 | 4 | 5 | 6 | 7 | 8 | 9 | 10 | 11 | 12 | 13 | Score | % |
|---|---|---|---|---|---|---|---|---|---|---|---|---|---|---|---|
| Adhikari 2020 [43] | 1 | 3 | 2 | 1 | 1 | 1 | 1 | 1 | 1 | 2 | 1 | 1 | 1 | 17 | 44 |
| Allison 2023 [44] | 3 | 3 | 3 | 3 | 3 | 3 | 3 | 3 | 3 | 3 | 3 | 3 | 2 | 38 | 97 |
| Anderseck 2020 [45] | 1 | 1 | 1 | 1 | 1 | 3 | 1 | 1 | 1 | 1 | 1 | 2 | 1 | 16 | 41 |
| Balogun 2018 [46] | 2 | 3 | 2 | 2 | 1 | 2 | 2 | 2 | 0 | 0 | 1 | 1 | 2 | 20 | 51 |
| Banks 2013 [47] | 3 | 3 | 3 | 3 | 2 | 2 | 1 | 2 | 2 | 2 | 3 | 3 | 3 | 32 | 82 |
| Barton 2021 [48] | 3 | 3 | 3 | 3 | 2 | 3 | 3 | 3 | 3 | 1 | 3 | 3 | 3 | 36 | 92 |
| Bastick 2020 [49] | 3 | 3 | 3 | 2 | 3 | 3 | 3 | 1 | 3 | 2 | 3 | 3 | 3 | 35 | 90 |
| Bird 2022 [50] | 3 | 3 | 3 | 3 | 3 | 3 | 3 | 3 | 1 | 3 | 3 | 2 | 2 | 35 | 90 |
| Brennan 2006 [51] | 3 | 3 | 3 | 3 | 2 | 3 | 3 | 3 | 3 | 3 | 3 | 2 | 3 | 37 | 95 |
| Briggs 2023 [52] | 3 | 3 | 3 | 2 | 3 | 3 | 3 | 3 | 3 | 3 | 3 | 3 | 3 | 38 | 97 |
| Camden 2015 [53] | 3 | 3 | 2 | 3 | 3 | 3 | 3 | 3 | 3 | 3 | 3 | 3 | 3 | 38 | 97 |
| Carr 2020 [54] | 3 | 3 | 2 | 3 | 3 | 3 | 3 | 3 | 2 | 3 | 3 | 3 | 1 | 35 | 90 |
| Cheema 2022 [55] | 3 | 3 | 2 | 2 | 3 | 2 | 2 | 3 | 3 | 3 | 3 | 2 | 2 | 33 | 85 |
| Chipchase 2016 [56] | 3 | 3 | 3 | 3 | 3 | 3 | 2 | 3 | 3 | 2 | 3 | 2 | 2 | 35 | 90 |
| Cimdi 2012 [57] | 1 | 2 | 3 | 2 | 2 | 2 | 1 | 3 | 1 | 0 | 1 | 1 | 1 | 20 | 51 |
| Cleland 2009 [58] | 3 | 3 | 2 | 3 | 0 | 2 | 3 | 3 | 2 | 3 | 3 | 0 | 2 | 29 | 74 |
| Constantine 2012 [59] | 3 | 3 | 3 | 3 | 3 | 3 | 3 | 3 | 3 | 3 | 3 | 3 | 3 | 39 | 100 |
| Cowell 2019 [60] | 3 | 3 | 3 | 2 | 3 | 2 | 3 | 3 | 3 | 2 | 3 | 3 | 2 | 35 | 90 |
| Cunningham 2017 [61] | 2 | 3 | 3 | 2 | 2 | 2 | 2 | 2 | 1 | 0 | 1 | 1 | 0 | 21 | 54 |
| Cunningham 2019 [62] | 3 | 3 | 3 | 3 | 2 | 3 | 3 | 2 | 1 | 2 | 3 | 2 | 2 | 32 | 82 |
| Cunningham 2021 [63] | 2 | 3 | 3 | 2 | 2 | 1 | 3 | 3 | 2 | 3 | 3 | 1 | 1 | 29 | 74 |
| Cunningham 2022 [64] | 3 | 3 | 3 | 3 | 2 | 3 | 3 | 3 | 1 | 3 | 3 | 2 | 3 | 35 | 90 |
| De Rooij 2020 [65] | 2 | 3 | 2 | 2 | 2 | 2 | 2 | 2 | 3 | 1 | 2 | 2 | 2 | 27 | 69 |
| Demmelmaier 2012 [66] | 3 | 2 | 3 | 2 | 1 | 2 | 2 | 3 | 2 | 2 | 2 | 2 | 3 | 29 | 74 |
| Dennis 1987 [67] | 2 | 2 | 3 | 3 | 3 | 3 | 3 | 1 | 2 | 0 | 2 | 0 | 1 | 25 | 64 |
| Deutscher 2014 [68] | 2 | 3 | 3 | 2 | 2 | 3 | 2 | 2 | 2 | 3 | 3 | 0 | 3 | 30 | 77 |
| Dizon 2014 [69] | 3 | 3 | 2 | 3 | 3 | 3 | 3 | 3 | 2 | 2 | 3 | 3 | 3 | 36 | 92 |
| Fary 2015 [70] | 3 | 3 | 3 | 2 | 3 | 3 | 3 | 3 | 3 | 3 | 3 | 3 | 3 | 38 | 97 |
| Furze 2019 [71] | 3 | 3 | 2 | 3 | 2 | 3 | 3 | 3 | 2 | 2 | 3 | 2 | 2 | 33 | 85 |
| Green 2008 [72] | 3 | 3 | 2 | 3 | 1 | 3 | 2 | 3 | 2 | 0 | 3 | 3 | 1 | 29 | 74 |
| Hansell 2023 [73] | 1 | 3 | 2 | 2 | 2 | 2 | 2 | 2 | 3 | 2 | 3 | 2 | 3 | 29 | 74 |
| Harrison 2022 [74] | 3 | 3 | 3 | 3 | 3 | 3 | 3 | 2 | 3 | 2 | 3 | 2 | 2 | 35 | 90 |
| Heneghan 2022 [75] | 3 | 3 | 3 | 3 | 2 | 3 | 3 | 3 | 3 | 3 | 3 | 3 | 2 | 37 | 95 |
| Jones 2008 [76] | 3 | 2 | 2 | 2 | 2 | 2 | 2 | 2 | 2 | 1 | 2 | 2 | 2 | 26 | 67 |
| Kafri 2023 [77] | 3 | 3 | 3 | 3 | 3 | 3 | 3 | 3 | 3 | 3 | 3 | 3 | 3 | 39 | 100 |
| Karas 2016 [78] | 3 | 2 | 2 | 2 | 1 | 1 | 2 | 1 | 1 | 0 | 2 | 2 | 3 | 22 | 56 |
| Karvonen 2015 [79] | 3 | 3 | 3 | 3 | 2 | 2 | 2 | 3 | 1 | 3 | 2 | 2 | 3 | 32 | 82 |
| Kerssens 1999 [80] | 2 | 2 | 2 | 0 | 0 | 1 | 1 | 2 | 1 | 1 | 1 | 1 | 2 | 16 | 41 |
| Lambrinos 2023 [81] | 3 | 3 | 3 | 3 | 3 | 3 | 2 | 3 | 3 | 3 | 3 | 3 | 2 | 37 | 95 |
| Lane 2022 [82] | 3 | 3 | 3 | 3 | 3 | 3 | 3 | 3 | 2 | 3 | 3 | 2 | 3 | 37 | 95 |
| Lawford 2018 [83] | 3 | 3 | 3 | 2 | 2 | 3 | 3 | 3 | 2 | 2 | 3 | 2 | 2 | 33 | 85 |
| Lawford 2019 [84] | 3 | 3 | 2 | 2 | 1 | 1 | 2 | 3 | 2 | 2 | 2 | 2 | 3 | 28 | 72 |
| Levsen 2001 [85] | 3 | 2 | 2 | 2 | 0 | 2 | 2 | 3 | 2 | 0 | 3 | 0 | 1 | 22 | 56 |
| Lonsdale 2017 [86] | 2 | 3 | 3 | 3 | 3 | 1 | 2 | 2 | 3 | 1 | 3 | 2 | 3 | 31 | 79 |
| Louw 2022 [87] | 3 | 3 | 2 | 3 | 2 | 2 | 3 | 3 | 1 | 1 | 3 | 2 | 2 | 30 | 77 |
| Maas 2012 [88] | 3 | 3 | 3 | 3 | 3 | 3 | 3 | 3 | 3 | 3 | 3 | 3 | 3 | 39 | 100 |
| MacPherson 2019 [89] | 3 | 3 | 2 | 3 | 2 | 1 | 3 | 2 | 3 | 1 | 2 | 3 | 2 | 30 | 77 |
| Madi 2018 [90] | 3 | 3 | 3 | 3 | 2 | 3 | 3 | 2 | 2 | 3 | 3 | 3 | 3 | 36 | 92 |
| Mansell 2020 [91] | 3 | 3 | 3 | 3 | 3 | 3 | 3 | 3 | 3 | 3 | 3 | 3 | 3 | 39 | 100 |

*(Continued)*

Table 3. (Continued)

| Author Year | 1 | 2 | 3 | 4 | 5 | 6 | 7 | 8 | 9 | 10 | 11 | 12 | 13 | Score | % |
|---|---|---|---|---|---|---|---|---|---|---|---|---|---|---|---|
| **March 2024 [92]** | 3 | 3 | 3 | 3 | 2 | 2 | 3 | 3 | 2 | 1 | 3 | 2 | 2 | 32 | 82 |
| **Murray 2015 [93]** | 3 | 3 | 2 | 3 | 3 | 2 | 3 | 3 | 2 | 3 | 3 | 3 | 3 | 36 | 92 |
| **Naidoo 2022 [94]** | 3 | 3 | 2 | 2 | 2 | 3 | 3 | 3 | 2 | 2 | 3 | 3 | 3 | 34 | 87 |
| **Ntoumenopoulos 2017 [95]** | 2 | 3 | 2 | 1 | 0 | 1 | 2 | 2 | 0 | 1 | 3 | 2 | 3 | 22 | 56 |
| **Olsen 2015 [96]** | 3 | 3 | 3 | 2 | 2 | 3 | 3 | 3 | 3 | 3 | 3 | 2 | 3 | 36 | 92 |
| **Overmeer 2009 [97]** | 3 | 2 | 2 | 2 | 2 | 3 | 3 | 3 | 2 | 1 | 2 | 2 | 2 | 29 | 74 |
| **Overmeer 2011 [98]** | 3 | 3 | 2 | 3 | 2 | 3 | 3 | 3 | 3 | 2 | 3 | 1 | 2 | 33 | 85 |
| **Perry 2011 [99]** | 3 | 3 | 2 | 2 | 2 | 2 | 3 | 3 | 2 | 2 | 3 | 3 | 1 | 31 | 79 |
| **Peter 2013 [100]** | 3 | 3 | 3 | 3 | 2 | 3 | 3 | 3 | 3 | 3 | 3 | 3 | 3 | 38 | 97 |
| **Peter 2015 [101]** | 2 | 3 | 1 | 2 | 2 | 2 | 2 | 3 | 3 | 2 | 3 | 3 | 3 | 31 | 79 |
| **Petty 2011 [102]** | 3 | 3 | 3 | 3 | 3 | 3 | 3 | 3 | 3 | 3 | 3 | 3 | 3 | 39 | 100 |
| **Petty 2011 [103]** | 3 | 3 | 3 | 3 | 3 | 2 | 3 | 3 | 3 | 3 | 3 | 3 | 2 | 37 | 95 |
| **Prizinski 2021 [104]** | 3 | 3 | 3 | 3 | 2 | 3 | 3 | 3 | 2 | 2 | 3 | 3 | 3 | 36 | 92 |
| **Rebbeck 2006 [105]** | 3 | 3 | 2 | 3 | 3 | 3 | 3 | 3 | 2 | 1 | 3 | 1 | 2 | 32 | 82 |
| **Resnik 2004 [106]** | 3 | 3 | 2 | 2 | 3 | 3 | 3 | 2 | 3 | 3 | 3 | 1 | 3 | 34 | 87 |
| **Rodeghero 2015 [107]** | 3 | 3 | 1 | 2 | 1 | 0 | 2 | 3 | 3 | 1 | 3 | 3 | 3 | 28 | 72 |
| **Rushton 2010 [108]** | 3 | 3 | 2 | 3 | 3 | 3 | 3 | 3 | 2 | 3 | 3 | 3 | 2 | 36 | 92 |
| **Schreiber 2012 [109]** | 3 | 3 | 2 | 2 | 2 | 3 | 3 | 3 | 3 | 2 | 2 | 2 | 3 | 33 | 85 |
| **Seif 2019 [110]** | 1 | 3 | 2 | 2 | 1 | 0 | 2 | 2 | 1 | 1 | 1 | 2 | 2 | 20 | 51 |
| **Shalabi 2024 [111]** | 3 | 3 | 3 | 2 | 3 | 2 | 2 | 2 | 2 | 2 | 2 | 1 | 3 | 30 | 77 |
| **SjodahlHammarlund 2013 [112]** | 3 | 3 | 2 | 3 | 2 | 3 | 3 | 2 | 2 | 2 | 3 | 3 | 2 | 33 | 85 |
| **Smith 1999 [113]** | 3 | 3 | 3 | 2 | 3 | 3 | 2 | 2 | 3 | 2 | 2 | 3 | 2 | 33 | 85 |
| **Souter 2019 [114]** | 3 | 3 | 2 | 2 | 2 | 3 | 2 | 3 | 3 | 3 | 2 | 3 | 3 | 34 | 87 |
| **Stathopoulos 2003 [115]** | 3 | 3 | 2 | 3 | 3 | 3 | 3 | 3 | 2 | 3 | 3 | 3 | 2 | 36 | 92 |
| **Stevenson 2006 [116]** | 3 | 3 | 3 | 2 | 2 | 2 | 2 | 3 | 1 | 3 | 3 | 3 | 2 | 32 | 82 |
| **Stevenson 2020 [117]** | 3 | 3 | 3 | 3 | 2 | 1 | 2 | 2 | 2 | 2 | 2 | 3 | 1 | 29 | 74 |
| **Swinkels 2015 [118]** | 3 | 3 | 3 | 2 | 2 | 1 | 1 | 3 | 3 | 1 | 2 | 1 | 2 | 27 | 69 |
| **Synnott 2016 [119]** | 3 | 3 | 2 | 3 | 3 | 2 | 3 | 3 | 3 | 2 | 3 | 2 | 3 | 35 | 90 |
| **Tilson 2014 [120]** | 3 | 3 | 3 | 3 | 3 | 3 | 3 | 3 | 3 | 2 | 3 | 3 | 3 | 38 | 97 |
| **Westervelt 2020 [121]** | 3 | 2 | 3 | 2 | 2 | 3 | 2 | 3 | 3 | 3 | 3 | 1 | 2 | 32 | 82 |
| **Whitman 2020 [122]** | 3 | 3 | 3 | 2 | 2 | 2 | 2 | 3 | 3 | 1 | 3 | 3 | 3 | 33 | 85 |
| **Williams 2019 [123]** | 3 | 3 | 3 | 3 | 3 | 3 | 3 | 3 | 2 | 3 | 3 | 2 | 3 | 37 | 95 |

Key for Table 3

1. Theoretical or conceptual underpinning to the research

2. Statement of research aim

3. Clear description of research setting and target population

4. Study design is appropriate to address stated research aims

5. Appropriate sampling to address the research aim

6. Rationale for choice of data collection tools

7. The format and content of data collection tool is appropriate to address the stated research aim

8. Description of data collection procedure

9. Recruitment data provided

10. Justification for analytic method selected

11. Method of analysis was appropriate to answer the research aim

12. Evidence that the research stakeholders have been considered in the design or conduct

13. Strengths and limitations critically discussed

These courses cross diverse areas of practice, but are often narrowly focused to an individual skill, concept or theory [53,73,81,82,95]. It is commonly comprised of didactic methods with components of interactive case studies and practical sessions. Distinct to this pathway is a lack of follow-up or carrying forward of learning over time, as the classification title suggests it tends to be a one and done occurrence. Like multiple encounter courses, inclusion of any formal assessment is rare as is accreditation and clinical mentorship [43,44,68,73,81,101].

**Pillars of Advanced Practice.** Table 4 and Fig 3a-f depict the results of the directed content analysis, respectively illustrating the consistency of pillar demonstration among studies and the frequency of competencies underpinning each pillar demonstrated in the PT after traversing each EP [34]. The latter is represented in the form of a heat map [36].

## Discussion

This SMSR aimed to understand and evaluate the PL-EPs that PTs engage in internationally to advance their level of practice through the lens of an established framework for Advanced Practice [1]. The main findings of this review support that there are 6 distinct PL-EPs that PTs engage in to advance their level of practice. These pathways are categorized as Masters level education, residency and fellowship programs, accredited area of practice education, mentorship, multiple encounter courses and single encounter courses. There is a high level of evidence to support that 3 of the 6 described pathways (Masters level, residency and fellowship and mentorship) can develop all 4 pillars of Advanced Practice in the PT. However, only Masters level education demonstrated these outcomes consistently with a moderate to high frequency of individual competency demonstration across all 4 pillars. There is low to moderate level of evidence to support that the remaining 3 described PL-EPs (accredited area of practice education, multiple encounter courses, single encounter courses) consistently identified PTs demonstrating only the clinical practice pillar, with no evidence of all 4 pillars in the PT following these PL-EPs.

There are both parallels and distinctions among the EPs described in this review. One commonality is the dominance of MSK, with 68% of the studies focused to this area of practice. This is coherent with the current global landscape, a 2024 international cross-sectional survey regarding APP identified that over half (53%) of survey respondents worked in the MSK area of practice [18]. Nonetheless, each defined pathway had unique identifiable components permitting its categorization. The key elements that contributed to distinguishing each pathway included the timeframe, the frequency of

**Table 4. Consistency of the pillars of Advanced Practice [1] demonstrated by the PT after each EP. Indicated by a percentage of the total number of studies in each category of EP.**

| Educational Pathway | Total number of studies | Clinical Practice | Leadership | Education | Research | All 4 pillars present | GRADE CERQual Level of Confidence |
|---|---|---|---|---|---|---|---|
| Masters Level Education [59,72,90,99,102,103,108,112,115] | 9 | 100% | 100% | 100% | 100% | 100% | High |
| Residency and Fellowship Programs [52,55,61–64,71,76,89,94,104,107,113,114,117,122] | 16 | 94% | 63% | 50% | 31% | 25% | High |
| Accredited Area of Practice Education [45,47,67,106] | 4 | 100% | 0% | 25% | 25% | 0% | Low |
| Mentorship [54,75,121,123] | 4 | 100% | 75% | 75% | 50% | 50% | High |
| Multiple Encounter Courses [49,51,58,60,65,66,70,74,79,80,83,85,88,91,96–98,109,110,118–120] | 22 | 95% | 18% | 14% | 18% | 0% | Moderate |
| Single Encounter Courses [43,44,46,48,50,53,56,57,68,69,73,77,78,81,82,84,86,87,92,93,95,100,101,105,111,116] | 26 | 92% | 8% | 4% | 12% | 0% | Moderate |

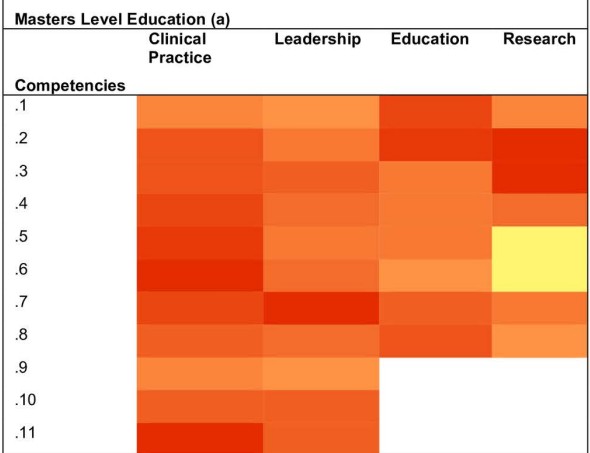

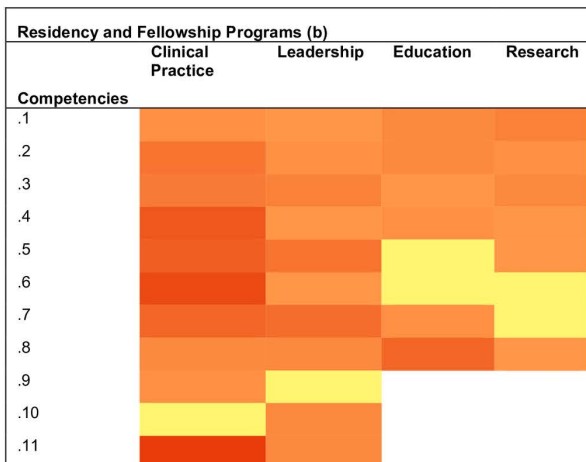

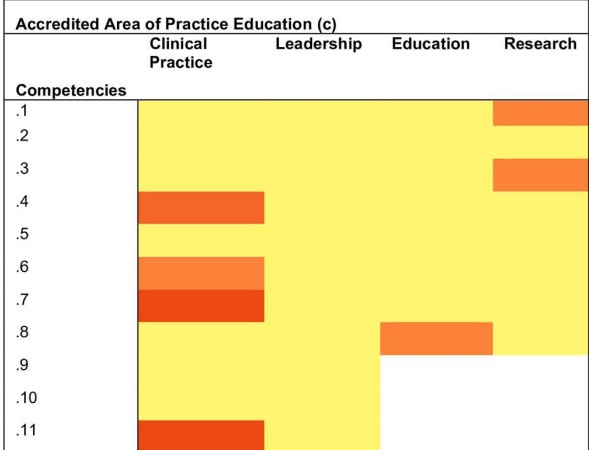

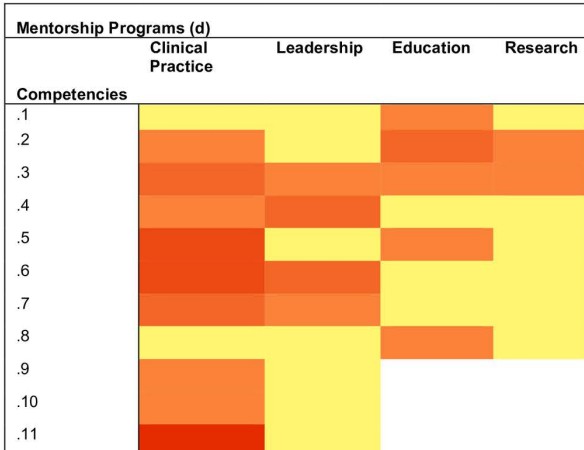

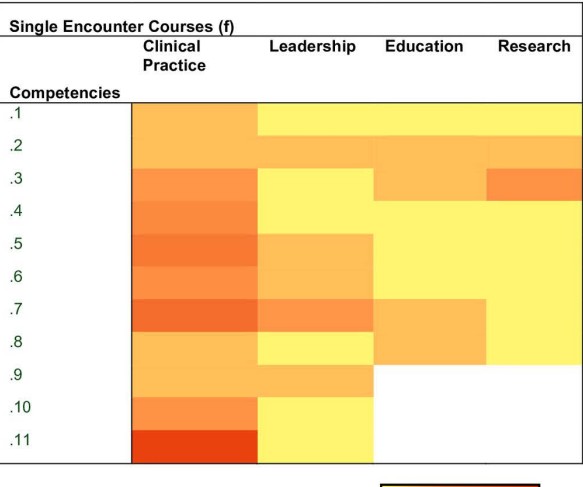

**Fig 3. a-f. Frequency of individual competency demonstration underpinning the pillars of Advanced Practice [ 1 ] demonstrated by the PT.**
Competencies coded most frequently are depicted in red, while those coded least frequently are in yellow. Descriptions of each competency (.1-.11) available in S6 Table.

engagement, accreditation status, inclusion of clinical mentorship, avenues for feedback, and requirement of formal student assessment. The clustered textual descriptions that arose from this synthesis are in line with previous literature that has characterized Masters level education, residency and fellowship programs as well as mentorship for the PT [15,125–128]. However, the classification of accredited area of practice education, multiple encounter courses and single encounter courses are novel to this review. Previous literature has defined educational interventions that encompass these groupings with varying terms such as professional development, continuing education and learning activities [129–132]. Until now, the common patterns within these PL-EPs have not been integrated in a way to allow for a wide lens evaluation of their role in the pathway to APP.

PL education for the PT has previously been appraised from a different angle. A recent set of sister quantitative and qualitative systematic reviews set out to understand PT's beliefs and perspectives regarding their experiences of PL education, and in parallel evaluate what was common about the experiences that enhanced their practice [130,131]. Interestingly, the PL-EPs which failed to evidence all 4 pillars of Advanced Practice in PTs in the present SMSR (single and multiple encounter courses), share many of the characteristics of 'learning activities' described by Leahy (2020) to be void of 'worthwhile learning', such as largely didactic engagement of limited frequency[130,131]. The PL-EPs that evidenced all 4 pillars shared extensive commonalities with the factors identified by Leahy (2020), as required for 'worthwhile' learning such as, reflection, mentorship, opportunity for feedback, and substantial time [130,131]. These reviews conclude that for PL education to be 'worthwhile' i.e. successfully impact the PT, they should include active approaches, such as peer assessment, mentored patient interactions, connected activities, time to practice, trustworthy resources and activities that take PTs out of their comfort zone. These conclusions are in line with the findings presented in this SMSR, and intriguingly, may explain the mechanisms by which the described PL-EPs successfully impacted the knowledge and behaviours of the PT such that embodiment of the pillars of Advanced Practice was detectable.

To our knowledge no previous study has looked to provide evidence of the pillars of Advanced Practice in PTs after traversing various PL-EPs. However, findings pertaining to this objective are largely in keeping with what previous literature has alluded to. In 2023, Peterson argued that while weekend continuing education courses (i.e., single and multiple encounter courses) can be an efficient way to refine or learn new skills, relying on them to improve knowledge and behavior towards an [advanced level of practice] is at best questionable [132]. A contrary finding in this SMSR to previous high-quality literature from the UK, is the lack of support for accredited area of practice education as a trustworthy and reliable pathway to APP [13]. The finding is supported here by a low level of confidence in evidence however and therefore should be interpreted with caution [37,39,40]. Other healthcare professions, particularly medicine and pharmacy have embraced a model of post-professional education in the form of residency and fellowship programs as well as ongoing mentorship across different areas of practice for many decades [127]. Program directors of these residency programs are asked to document how their residents achieve proficiency in 8 domains of competence that in keeping with the findings of this SMSR, map to the clinical practice, leadership and education pillars, but are void of the research pillar [127,133]. Reiterating that while these PL-EPs do well to cultivate competencies aligned with 3 of the 4 pillars, they fall short of providing a comprehensive framework.

Ultimately, Masters level education was the only pathway to consistently show PTs demonstrating all 4 pillars of Advanced Practice. Considering the required pathway to Advanced Practice in other healthcare professions such as nursing, these results are not surprising, nor is the existing literature to support the impact Masters level education has for nurses in all 4 pillars [109,12,134]. Multiple studies have previously supported Masters level education as a critical stop on the journey to developing clinical expertise for PTs, which is also supported here by the breadth and depth of demonstration of the clinical practice pillar [125,126,135]. However, the critical findings in this review support that Masters level education is likely the optimal pathway to developing the remaining 3 pillars of Advanced Practice as well, a fact that until now has largely been supported by observation, anecdotal evidence and expert opinion [1,11,18]. The high level of confidence in the finding that Masters level education is the most consistent pathway to embodiment of the 4 pillars of Advanced Practice for the PT is a critical step toward firmly establishing APP in healthcare systems internationally.

**Strengths and Limitations**

This SMSR was novel in its exploration of the potential PL-EPs to APP for the PT through the lens of the well-established pillars of Advanced Practice [1]. Rigorous methods were employed at all stages, including a robust search strategy, a complete parallel review approach [136], the use of valid and reliable research tools [28], including assessment of the confidence in evidence supporting the review findings using the GRADE CERQual [137] and lastly the inclusion of a positionality statement of the researchers to enable readers to interpret the findings and judge trustworthiness in their own context [138,139]. Despite these strengths, this review is not without limitations. Studies included were conducted in 5 of 7 continents. While the results are largely representative of EPs engaged in by the PT internationally, applicability of the findings across countries should be considered with caution as global differences in healthcare and education systems may limit generalizability. Additionally, this SMSR facilitated a comprehensive synthesis of diverse evidence but required a pragmatic integrative approach to accommodate studies with varied aims and heterogeneous data sets. The process of qualitizing quantitative data for synthesis introduced inherent challenges, particularly regarding the depth and nuance embedded in qualitative data. While this approach allowed for a unified analytical framework, the transformation of numerical findings into qualitative themes inevitably involved interpretive decisions by the researchers. These decisions, shaped by the coding framework and analytical lens, may have influenced how meaning was constructed from the data, with potential implications for the richness and contextual depth of the synthesized findings. Thus, while this method enabled a more cohesive synthesis, it is important to acknowledge that some intricacies of the original quantitative data may have been lost in translation, highlighting a key consideration when drawing conclusions from qualitized data [31,38,28,139].

**Research Implications.** There is an opportunity to further characterize the effect PL Masters level education has on PTs. Particularly, well-designed prospective cohort studies would be of benefit to gain greater insight into the magnitude of the effect and the mechanisms by which this PL-EP impacts the PT. This research is particularly needed in countries such as Canada, where APP is on the forefront of change for healthcare systems and PL Masters level educational programs specific to APP exist [19]. The aim of healthcare research is to better the lives of the end-user, the patient. As such, the current findings should be taken forward in the way of implementation science, to investigate how PTs with PL Masters level education that embody the 4 pillars of Advanced Practice, impact patients and healthcare systems. Lastly, there is a need for studies of high methodological quality investigating accredited area of practice education to improve the certainty of evidence of this pathway as it lends to APP.

## Conclusions

This review highlights 6 distinct PL-EPs that PTs pursue to advance their practice. These EPs are categorized as Masters level education, residency and fellowship programs, accredited area of practice education, mentorship, multiple encounter courses, and single encounter courses. High level of confidence in the evidence supports the ability of 3 of these pathways—Masters level education, residency and fellowship programs, and mentorship, to foster competencies across all 4 pillars of Advanced Practice. Masters level education is the only pathway that consistently achieved these outcomes with a moderate to high frequency of individual competency demonstration across all 4 pillars. There is low to moderate confidence in the evidence suggesting that the remaining 3 pathways—accredited area of practice education, multiple encounter courses, and single encounter courses—primarily support the clinical practice pillar, with no evidence of comprehensive competency development across all 4 pillars.

## Supporting information

**S1 Table. Completed PRISMA-P checklist.**
(DOCX)

**S2 Table. Adapted search strategies.**
(DOCX)

**S3 Table. Objective 1 data extraction and coding.**
(DOCX)

**S4 Table. Objective 2 data extraction and coding.**
(DOCX)

**S5 Table. GRADE-CERQual evidence profile.**
(DOCX)

**S6 Table. Advanced Practice pillar competencies.**
(DOCX)

**S7 Table. List of all studies identified in literature search.**
(DOCX)

## Acknowledgments

Christy Sich, Teaching and Learning Librarian at Western University (London, Ontario, Canada) for support with reviewing and shaping the search strategy.

## Author contributions

**Conceptualization:** Kaitlyn Maddigan, Katie L Kowalski, Alison B Rushton.

**Data curation:** Kaitlyn Maddigan, Chris Davis, Brendan Saville, Kathryn Nishimura.

**Formal analysis:** Kaitlyn Maddigan, Chris Davis.

**Investigation:** Kaitlyn Maddigan.

**Methodology:** Kaitlyn Maddigan, Chris Davis, Brendan Saville, Andrews K Tawiah, Katie L Kowalski, Alison B Rushton.

**Project administration:** Kaitlyn Maddigan.

**Resources:** Kaitlyn Maddigan.

**Software:** Kaitlyn Maddigan.

**Supervision:** Jennifer Van Bussel, Andrews K Tawiah, Katie L Kowalski, Alison B Rushton.

**Visualization:** Kaitlyn Maddigan, Andrews K Tawiah, Katie L Kowalski, Alison B Rushton.

**Writing – original draft:** Kaitlyn Maddigan.

**Writing – review & editing:** Kaitlyn Maddigan, Chris Davis, Brendan Saville, Jennifer Van Bussel, Andrews K Tawiah, Katie L Kowalski, Alison B Rushton.

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
