## [Decision Letter · Decision Letter 0]

25 Feb 2025

PONE-D-24-58918The educational pathway to Advanced Practice for the physiotherapist: a systematic mixed studies reviewPLOS ONE

Dear Dr. Maddigan,

Thank you for submitting your manuscript to PLOS ONE. After careful consideration, we feel that it has merit but does not fully meet PLOS ONE’s publication criteria as it currently stands. Therefore, we invite you to submit a revised version of the manuscript that addresses the points raised during the review process.

Reviewer #1: Thank you for the opportunity to review this paper. The aim of the paper was to undertake a systematic review of the educational requirements for to advanced physiotherapy practice. This is clearly set out in the aims of the study. The review has been registered with prospero and the methodology for the review is clearly set out. The search has revealed a wide range of relevant papers and these have been graded for their quality. The results and clearly outlined in the relevant tables and figures and the discussion and conclusions are based clearly on the results. The paper is well written and adds valuable knowledge to the topic. I think it will be a useful review and used in the profession to guide future education standards for APPs across the world. I have no other comments to add to the paper.

Reviewer #2: I thank the authors for the opportunity to review their paper. This article is a systematic mixed studies review. The study aims to describe the post-licensure educational pathways that physiotherapists engage in to advance their level of practice and to evaluate the pillars of APP demonstrated by the physiotherapist after traversing a post-licensure educational pathway. This study is very interesting and undeniably useful regarding the implementation of advanced practice in physiotherapy. The methodology used is ambitious and appropriate, although the analysis method lacks precision.

Overall, the paper is written well. Below are some specific comments to take into consideration.

Abstract:

The different sections of the abstract (background, methods, results, and discussion) could be more clearly delineated to enhance readability and improve the overall structure.

The total number of screened studies does not need to be included in the abstract; however, the study design of the 81 included studies should be specified in the abstract.

Line 35: “Masters level education is the optimal pathway to APP”: The last sentence of the conclusion should be nuanced in light of the methodology used (mixed review and thematic content analysis). See comment below, in the discussion section.

Introduction:

Line 40 to 50: It could be valuable to provide examples of the clinical practice areas of advanced physiotherapy practice and the roles of advanced practice physiotherapists, as this is an emerging role in some countries. This would help readers gain a clearer understanding of its practical application across different healthcare systems.

Line 51: Reference 2 appears in the middle of the sentence, which disrupts readability. References should systematically be placed at the end of sentences throughout the manuscript.

Line 60 to 64: The sentence is long and difficult to read, making its meaning unclear. It should be reworded for better clarity.

Line 67 to 69: This statement should be nuanced, as the lack of standardization in advanced practice education is not the only factor slowing down the implementation and integration of these roles in healthcare systems worldwide.

Objectives:

Line 81: I am not sure that the term "evaluate" is the most appropriate in the formulation of the second objective, given the study design used to address this objective. Isn't it more about exploring the representativeness of the pillars of advanced practice within the different training programs, or determining to what extent these pillars are or are not represented in each curriculum? The term "evaluate" refers to a quantitative design based on objective outcomes.

Methods:

Table 1: Please clarify whether outcomes 1 and 2 described in the eligibility criteria table should be combined or can be considered separately.

Table 1: I suggest justifying the inclusion of qualitative, quantitative, and mixed designs in light of the stated objectives.

Line 101: Please clarify whether publication in a peer-reviewed journal was an inclusion criterion. If not, please justify the reason.

Line 116 to 122: Even though the search equations are available in the supplementary material, it would be helpful to specify the main keywords/MeSH terms used for the search.

Line 126 to 129: Please specify the method used in case of disagreement between two researchers regarding the study selection, and identify the researchers involved in this step.

Line 156 to 163: The method for analysing mixed systematic reviews is relatively unknown and presents a challenge given the quantitative and qualitative nature of the collected data. The analysis plan should therefore be more detailed, particularly to specify how quantitative data are "transformed" into qualitative data (or “qualitized”).

Additionally, please clarify which specific method is used to address each of the two objectives. For a reader who is not familiar with this type of analysis, the current description (both text and Figure 2) appears to lack sufficient detail.

Results:

Line 201 to 207: It is not clear in this section that the GRADE-CERQual assessment of the level of evidence applies only to qualitative study designs, and not to all included studies. I suggest clarifying this to help the reader better understand the simultaneous use of two quality assessment tools for the included studies.

Line 226 to 231: Please clarify whether all of these programs are focused on post-graduate training, and thus accessible to already qualified physiotherapists, or if some programs are linked to the initial training of physiotherapists.

Line 233 to 286: The specifics of each program are well presented, and the summary is very clear. Would it be possible to add the studies that reference each program characteristic mentioned within each paragraph?

Table 4: Please specify in Table 4 which studies the GRADE-CERQual evidence level rating refers to, among the studies cited, given that this tool applies only to qualitative study designs.

Discussion:

General comment: The discussion section is interesting but quite lengthy, with some parts being repetitive. I suggest shortening it slightly to emphasize the key points and make the reading of this section more concise and easier to follow.

Line 311 to 313: Given the heterogeneity of the included study designs and the use of a qualitative analysis method applied to quantitative studies, it would seem appropriate to nuance the statement that the master's degree is the only one that fully embodies the four pillars of advanced practice. Perhaps the content analysis did not allow for this to be highlighted for the other curricula mentioned, as this analysis method is not intended to systematically assess an outcome, but rather to explore its various dimensions.

Line 424 to 426: “Specifically, it is inherently harder to harness the detail and intricacies in transformed quantitative data that is naturally offered in qualitative data sets.”: Please provide more detail on the implications of this methodological limitation on the results obtained.

-----------------

We look forward to receiving your revised manuscript.

Kind regards,

Anthony Demont

Academic Editor

PLOS ONE

2. We note that your Data Availability Statement is currently as follows: [All relevant data are within the manuscript and its Supporting Information files.]Please confirm at this time whether or not your submission contains all raw data required to replicate the results of your study. Authors must share the “minimal data set” for their submission. PLOS defines the minimal data set to consist of the data required to replicate all study findings reported in the article, as well as related metadata and methods (https://journals.plos.org/plosone/s/data-availability#loc-minimal-data-set-definition).

3. As required by our policy on Data Availability, please ensure your manuscript or supplementary information includes the following:

Reviewer's Responses to Questions

**Comments to the Author**

1. Is the manuscript technically sound, and do the data support the conclusions?

Reviewer #1: Yes

Reviewer #2: Yes

2. Has the statistical analysis been performed appropriately and rigorously? 

Reviewer #1: Yes

Reviewer #2: N/A

3. Have the authors made all data underlying the findings in their manuscript fully available?

Reviewer #1: Yes

Reviewer #2: Yes

4. Is the manuscript presented in an intelligible fashion and written in standard English?

Reviewer #1: Yes

Reviewer #2: Yes

5. Review Comments to the Author

6. PLOS authors have the option to publish the peer review history of their article (what does this mean? ). If published, this will include your full peer review and any attached files.

**Do you want your identity to be public for this peer review?** For information about this choice, including consent withdrawal, please see our Privacy Policy .

Reviewer #1: **Yes: ** Duncan Reid

Reviewer #2: No

---

## [Author Response · Author response to Decision Letter 1]

18 Mar 2025

Response to Reviewers

Thank you to reviewers 1 and 2 for their review and comments.

Reviewer 1 Comment:

Thank you for the opportunity to review this paper. The aim of the paper was to undertake a systematic review of the educational requirements for to advanced physiotherapy practice. This is clearly set out in the aims of the study. The review has been registered with prospero and the methodology for the review is clearly set out. The search has revealed a wide range of relevant papers and these have been graded for their quality. The results and clearly outlined in the relevant tables and figures and the discussion and conclusions are based clearly on the results. The paper is well written and adds valuable knowledge to the topic. I think it will be a useful review and used in the profession to guide future education standards for APPs across the world. I have no other comments to add to the paper.

Response: thank you for taking the time to review this manuscript. Your encouraging words pertaining to the quality of the work are greatly appreciated, as is your endorsement for publication.

Reviewer 2 Comment:

I thank the authors for the opportunity to review their paper. This article is a systematic mixed studies review. The study aims to describe the post-licensure educational pathways that physiotherapists engage in to advance their level of practice and to evaluate the pillars of APP demonstrated by the physiotherapist after traversing a post-licensure educational pathway. This study is very interesting and undeniably useful regarding the implementation of advanced practice in physiotherapy. The methodology used is ambitious and appropriate, although the analysis method lacks precision. Overall, the paper is written well. Below are some specific comments to take into consideration.

Response: thank you for taking the time to review the manuscript. Thank you for your endorsement in the appropriateness of the methodology and for your suggestions to improve the quality of the paper, they are greatly appreciated. See below for a detailed account of the revisions that have been carried out.

Response to reviewer 2’s specific comments:

Abstract

1. The different sections of the abstract (background, methods, results, and discussion) could be more clearly delineated to enhance readability and improve the overall structure.

• This change was made - the subheadings: background, objectives, methods, results and conclusion were added to the abstract. Please see lines 34-53.

2. The total number of screened studies does not need to be included in the abstract; however, the study design of the 81 included studies should be specified in the abstract.

• The total number of studies was removed from the abstract on line 30. The number of included studies as well as the study designs are included on line 35.

3. Line 35: “Masters level education is the optimal pathway to APP”: The last sentence of the conclusion should be nuanced in light of the methodology used (mixed review and thematic content analysis). See comment below, in the discussion section.

• We agree that this sentence could be nuanced in light of the methodology and was revised on line 53 to reflect “Masters level education appears to be the optimal pathway.”

Introduction

4. Line 40 to 50: It could be valuable to provide examples of the clinical practice areas of advanced physiotherapy practice and the roles of advanced practice physiotherapists, as this is an emerging role in some countries. This would help readers gain a clearer understanding of its practical application across different healthcare systems.

• The following sentences are now included in the introduction on lines 67-73: ‘APPs are most frequently found in musculoskeletal (MSK) care (including outpatient orthopaedics and sports physiotherapy), neurology, cardiorespiratory, and paediatrics. Their roles vary by specialty and region but often include requesting diagnostic imaging, ordering blood tests, performing injections, and independently prescribing or de-prescribing medications. APPs also conduct orthopaedic triage, screening patients in emergency departments and those referred for surgical consultation’.

5. Line 51: Reference 2 appears in the middle of the sentence, which disrupts readability. References should systematically be placed at the end of sentences throughout the manuscript.

• The reference was deleted from the middle of the sentence and included at the end.

6. Line 60 to 64: The sentence is long and difficult to read, making its meaning unclear. It should be reworded for better clarity.

• This sentence was rephrased on lines 89-91 to: “Moreover, a recent scoping review on APP examined education curricula and advocated for uniform standards, emphasizing that despite variations in APP roles within and across countries, standardized education remains feasible.”

7. Line 67 to 69: This statement should be nuanced, as the lack of standardization in advanced practice education is not the only factor slowing down the implementation and integration of these roles in healthcare systems worldwide.

• This sentence was rephrased on lines 94-96 to: “Lack of standardized PL-EPs is a key factor that contributes to the slow acknowledgment, growth and integration of these roles into healthcare systems worldwide.”

8. Line 81: I am not sure that the term "evaluate" is the most appropriate in the formulation of the second objective, given the study design used to address this objective. Isn't it more about exploring the representativeness of the pillars of advanced practice within the different training programs, or determining to what extent these pillars are or are not represented in each curriculum? The term "evaluate" refers to a quantitative design based on objective outcomes.

• We do not disagree entirely with the interpretation presented here but propose that ‘evaluate’ can also be terminology used across mixed study designs. Therefore, we have re-arranged the words to reflect the objective more clearly. See line 120-121: “To evaluate demonstration of the pillars of Advanced Practice by the PT after traversing a PL-EP.”

Methods

9. Table 1: Please clarify whether outcomes 1 and 2 described in the eligibility criteria table should be combined or can be considered separately.

• Outcomes 1 and 2 are intended to be considered separately, identifying each by numbers 1 and 2 in the table was intended to indicate that they should be considered in this way.

10. Table 1: I suggest justifying the inclusion of qualitative, quantitative, and mixed designs in light of the stated objectives.

• This justification is included in the section on study design just above the table on lines 131-139.

11. Line 101: Please clarify whether publication in a peer-reviewed journal was an inclusion criterion. If not, please justify the reason.

• Publication in a peer-reviewed journal was not a requirement for inclusion in this SMSR. Instead, studies had to be primary research, as outlined in Table 1. Grey literature was intentionally included in the information sources as outlined in lines 150-153, due to the novelty of the topic and the expected scarcity of primary research in education. This approach allowed for the inclusion of all relevant studies that met the criteria for primary research. Transparency of reporting illustrates the nature of each included study.

12. Line 116 to 122: Even though the search equations are available in the supplementary material, it would be helpful to specify the main keywords/MeSH terms used for the search.

• A brief list of MeSH terms has been added to the search strategy section on lines 162-167.

13. Line 126 to 129: Please specify the method used in case of disagreement between two researchers regarding the study selection, and identify the researchers involved in this step.

• These details have been added with the researchers involved in study selection on line 172 and the method used in case of disagreement on lines 178-180.

14. Line 156 to 163: The method for analysing mixed systematic reviews is relatively unknown and presents a challenge given the quantitative and qualitative nature of the collected data. The analysis plan should therefore be more detailed, particularly to specify how quantitative data are "transformed" into qualitative data (or “qualitized”). Additionally, please clarify which specific method is used to address each of the two objectives. For a reader who is not familiar with this type of analysis, the current description (both text and Figure 1) appears to lack sufficient detail.

• A more detailed explanation of both the data transformation process as well as the intricacies of the qualitative synthesis approach is now included in the Data Synthesis section, see lines 211-224.

• More detail has also been added to figure 1 and uploaded separately.

Results

15. Line 201 to 207: It is not clear in this section that the GRADE-CERQual assessment of the level of evidence applies only to qualitative study designs, and not to all included studies. I suggest clarifying this to help the reader better understand the simultaneous use of two quality assessment tools for the included studies.

• The use of the QUADS was operationalized at a study level, first to inform the reader of the quality of each individual study included and subsequently to inform the application of CERQual at the level of synthesis across studies, for the studies that contributed to each individual finding. This is described on lines 233-238.

• Mixing qualitative and quantitative evidence occurs at two levels: study and synthesis. GRADE-CERQual applies only to the synthesis level, not exclusively to qualitative study designs. Thus, all studies contributing to the synthesized review findings were assessed using GRADE-CERQual. As Lewin et al. (2018) describe, GRADE-CERQual evaluates review findings from a qualitative evidence synthesis, defined as an analytic output describing a phenomenon based on primary study data. Since GRADE-CERQual does not require primary research to be qualitative and the synthesis in this review followed data transformation, its use to assess confidence in the cumulative qualitative evidence synthesis is justified[1].

• A brief justification of this nature has been added to the methods on lines 228-232.

• Finally, revisions were made to better reflect that GRADE-CERQual was applied to review findings and not at the study level in the results section as justified above. See lines 287-289: “12 review findings, summated from the synthesized primary data were assessed using the GRADE-CERQual. The assessment of cumulative evidence determined there to be high confidence in 6/12, moderate confidence in 4/12 and low confidence in 2/12 review findings”

16. Line 226 to 231: Please clarify whether all of these programs are focused on post-graduate training, and thus accessible to already qualified physiotherapists, or if some programs are linked to the initial training of physiotherapists.

• The term adopted throughout this review: post-licensure (PL) is used to signify that each of these pathways are focused to post-graduate training. This was also identified in the eligibility criteria as participants had to be physiotherapists, and the education was specified as post-licensure.

17. Line 233 to 286: The specifics of each program are well presented, and the summary is very clear. Would it be possible to add the studies that reference each program characteristic mentioned within each paragraph?

• Reference to specific studies have been included in the synthesized textual descriptions of the PL-EPs. See lines 322-378.

18. Table 4: Please specify in Table 4 which studies the GRADE-CERQual evidence level rating refers to, among the studies cited, given that this tool applies only to qualitative study designs.

• Each of the studies referenced in Table 4 contributed data to the synthesized findings that underwent assessment using the GRADE CERQual.

• See response to comment #15.

19. General comment: The discussion section is interesting but quite lengthy, with some parts being repetitive. I suggest shortening it slightly to emphasize the key points and make the reading of this section more concise and easier to follow.

• Changes have been made to reduce redundancy in the discussion session. The total word count was reduced from 1263 to 1051. These changes are noted throughout lines 397-556.

20. Line 311 to 313: Given the heterogeneity of the included study designs and the use of a qualitative analysis method applied to quantitative studies, it would seem appropriate to nuance the statement that the master's degree is the only one that fully embodies the four pillars of advanced practice. Perhaps the content analysis did not allow for this to be highlighted for the other curricula mentioned, as this analysis method is not intended to systematically assess an outcome, but rather to explore its various dimensions.

• This sentence is purposefully written to highlight that Masters level education was the only PL-EP to consistently have PTs demonstrate all four pillars, not the only PL-EP in the review that had evidence of all four pillar embodiment in the PTs. This sentence is now on line 405-407 and reads: “However, only Masters level education demonstrated these outcomes consistently with a moderate to high frequency of individual competency demonstration across all four pillars.”

• Additionally, in response to the suspected limitations of a content analysis to systematically assess an outcome, as stated by Hseish and Shannon 2005, qualitative content analysis is defined as a method that employs a systematic process of coding, that is both logical and scientific[2]. The development of a good coding scheme is central to trustworthiness in a content analysis. This was fundamental to the present review which used an existing framework for Advanced Practice to code all collected data in a rigorous and systematic fashion. Going beyond simply counting words or numbers to examining the data for the purpose of categorizing information that represent similar meanings whether explicit or inferred through data transformation.

21. Line 424 to 426: “Specifically, it is inherently harder to harness the detail and intricacies in transformed quantitative data that is naturally offered in qualitative data sets.”: Please provide more detail on the implications of this methodological limitation on the results obtained.

• More detail has been provided pertaining to the limitations of the synthesis methods, see lines: 613-625.

Sincerely,

Kaitlyn Maddigan, on behalf of all authors

School of Physiotherapy, Western University, London, ON

kmaddig@uwo.ca

References

[1] Lewin S, Booth A, Glenton C, Munthe-Kaas H, Rashidian A, Wainwright M, et al. Applying GRADE-CERQual to qualitative evidence synthesis findings: introduction to the series. Implementation Science 2018;13:2. https://doi.org/10.1186/s13012-017-0688-3.

[2] Hsieh H-F, Shannon SE. Three Approaches to Qualitative Content Analysis. Qual Health Res 2005;15:1277–88. https://doi.org/10.1177/1049732305276687.

---

## [Editor Report · Decision Letter 1]

25 Mar 2025

The educational pathway to Advanced Practice for the physiotherapist: a systematic mixed studies review

PONE-D-24-58918R1

Dear Dr. Maddigan,

We’re pleased to inform you that your manuscript has been judged scientifically suitable for publication and will be formally accepted for publication once it meets all outstanding technical requirements.

Kind regards,

Anthony Demont

Academic Editor

PLOS ONE

---

## [Editor Report · Acceptance letter]

PONE-D-24-58918R1

PLOS ONE

Dear Dr. Maddigan,

I'm pleased to inform you that your manuscript has been deemed suitable for publication in PLOS ONE. Congratulations! Your manuscript is now being handed over to our production team.

Kind regards,

on behalf of

Dr. Anthony Demont

Academic Editor

PLOS ONE